# Antimicrobial Peptide Delivery Systems as Promising Tools Against Resistant Bacterial Infections

**DOI:** 10.3390/antibiotics13111042

**Published:** 2024-11-04

**Authors:** Kamila Botelho Sampaio de Oliveira, Michel Lopes Leite, Nadielle Tamires Moreira Melo, Letícia Ferreira Lima, Talita Cristina Queiroz Barbosa, Nathalia Lira Carmo, Douglas Afonso Bittencourt Melo, Hugo Costa Paes, Octávio Luiz Franco

**Affiliations:** 1Centro de Análises Proteômicas e Bioquímicas, Pós-Graduação em Ciências Genômicas e Biotecnologia, Universidade Católica de Brasília, Federal District, Brasilia 71966-700, Brazil; kamilasampaio18@gmail.com (K.B.S.d.O.); nadytamires@gmail.com (N.T.M.M.); lethicialima55@gmail.com (L.F.L.); tataqueiroz22@gmail.com (T.C.Q.B.); nathalialc41@gmail.com (N.L.C.); douglas.am@sempreceub.com (D.A.B.M.); 2S-Inova Biotech, Pós-Graduação em Biotecnologia, Universidade Católica Dom Bosco, Campo Grande 79117-900, Brazil; 3Departamento de Biologia Molecular, Instituto de Ciências Biológicas, Campus Darcy Ribeiro, Bloco K, Universidade de Brasília, Federal District, Brasilia 70790-900, Brazil; michelleitte@gmail.com; 4Grupo de Engenharia de Biocatalisadores, Faculdade de Medicina, Campus Darcy Ribeiro, Universidade de Brasília, Federal District, Brasilia 70790-900, Brazil; sorumbatico@gmail.com; 5Divisão de Clínica Médica, Faculdade de Medicina, Campus Darcy Ribeiro, Universidade de Brasília, Federal District, Brasilia 70910-900, Brazil; 6Pós-Graduação em Patologia Molecular, Campus Darcy Ribeiro, Universidade de Brasília, Brasilia 70790-900, Brazil

**Keywords:** bacterial resistance, antimicrobial peptides, nanoparticles, nanomaterials, delivery systems

## Abstract

The extensive use of antibiotics during recent years has led to antimicrobial resistance development, a significant threat to global public health. It is estimated that around 1.27 million people died worldwide in 2019 due to infectious diseases caused by antibiotic-resistant microorganisms, according to the WHO. It is estimated that 700,000 people die each year worldwide, which is expected to rise to 10 million by 2050. Therefore, new and efficient antimicrobials against resistant pathogenic bacteria are urgently needed. Antimicrobial peptides (AMPs) present a broad spectrum of antibacterial effects and are considered potential tools for developing novel therapies to combat resistant infections. However, their clinical application is currently limited due to instability, low selectivity, toxicity, and limited bioavailability, resulting in a narrow therapeutic window. Here we describe an overview of the clinical application of AMPs against resistant bacterial infections through nanoformulation. It evaluates metal, polymeric, and lipid AMP delivery systems as promising for the treatment of resistant bacterial infections, offering a potential solution to the aforementioned limitations.

## 1. Introduction

Antimicrobial resistance (AMR) is a huge threat to public health today [1]. It is estimated that infections caused by bacteria resistant to the main classes of antibiotics caused around 4.95 million deaths in 2019 alone, with low and medium-income countries being the most affected, particularly in sub-Saharan Africa [2]. Every year, approximately 700,000 deaths around the world occur due to infections caused by multi-resistant bacteria [3]. It is estimated that by 2050, globally, the number of deaths will reach around 10 million each year, and the costs of treating these infections will total around 100 trillion dollars [4].

Therefore, advanced therapeutic approaches must be employed to combat these resistant microorganisms [5]. One approach could be antimicrobial peptides (AMPs) as an alternative to current antibiotics to combat resistant infections [6,7]. AMPs are biochemically conserved molecules produced by virtually all living organisms as the first line of defense against pathogen invasion [8], including several Gram-positive and -negative harmful bacteria [9,10]. They are oligopeptides (five to 100 amino acids) with a positive net charge (+2 to +11), approximately 50% of hydrophobic residues in their composition, and show a broad spectrum of target microorganisms [11]. AMPs can also act as immunomodulators of the inflammatory response by stimulating the proliferation and recruitment of macrophages, neutrophils, eosinophils, and T lymphocytes [12,13]. AMPs can combat bacteria through various mechanisms, acting on the cell wall of microbial cells, affecting the synthesis of essential components such as peptidoglycans, teichoic acid, lipoproteins, and lipopolysaccharide, and destroying its structure. They can also affect the process of bacterial division [5,14]. AMPs interact with bacterial cells through electrostatic interactions and gradually accumulate on the cell membrane surface until a threshold concentration is reached. After that, AMPs act through different modes of action, forming transmembrane channels involving rod pores or toroidal pores on the target cell membrane [5,15]. They can also act through the formation of a carpet that covers the membrane surface and causes its disintegration to form micelles, a process also known as the detergent-type model or by the so-called Shai–Huang–Matsuzaki (SHM) model, whereby the peptides act through different mechanisms [16,17,18]. These mechanisms of action increase membrane permeability, leading to leakage of cell contents and cell death [14]. Furthermore, AMPs can cross the cell wall through direct penetration or endocytosis and inhibit the synthesis or activity of intracellular molecules, such as nucleic acids and intracellular proteins [19].

Despite their excellent antimicrobial properties, the clinical use of AMPs is still challenging due to some drawbacks. AMPs usually have limited stability, bioavailability, membrane permeability, half-life, and rapid clearance under physiological conditions. Different salt concentrations and pH values, or interaction with serum molecules such as the proteases [20,21], can compromise the in vivo activity of AMPs [22,23]. AMPs also present challenges related to their production, such as high cost and large-scale production [24].

Bioavailability is significant for developing peptide-based drugs, as it is critical for achieving the expected pharmaceutical efficacy with a minimal dose. This reduces toxicity and side effects, increasing the selectivity of the drugs [20]. However, using conventional administration routes for delivery of AMPs causes their rapid elimination, which means that doses are not maintained within the therapeutic window. Cytotoxic potential and stability problems due to enzymatic degradation and pH changes compound the problem [20]. Another relevant problem related to AMPs is that some bacteria are already resistant to AMPs due to natural selection. Resistant bacteria can cause modifications in components of the membrane structure, neutralize AMPs through secreted proteases, or expel transmembrane AMPs through efflux pumps. There are already marketed AMP-based drugs that present disadvantageous, mainly considerable adverse reactions, which still limit their complete clinical application [25]. The commercial AMPs vancomycin [26] and murepavadin [27] have strong antibacterial activity but also present ototoxicity, nephrotoxicity, allergy, diarrhea, oral intolerance, inflammation, and renal toxicity as limitations [25].

To overcome these disadvantages, ways must be developed to help peptide drugs cross biological barriers (e.g., gastric fluid acidity and peptidases, as well as poor membrane permeability due to their size and hydrophobicity) in vivo and reach the bloodstream for further distribution to their site of action [20]. This calls for further research and development of efficient AMP formulations.

Among the alternatives to improve the stability and bioavailability of peptide drugs, delivery systems based on AMP nanoencapsulation stand out. These may prove essential to bring AMPs to clinical application in treating systemic and local infection [28,29]. Nanomaterials have great potential to improve AMP pharmacokinetics and pharmacodynamics, protecting these molecules against serum proteases, preserving their activity, and minimizing side effects [23]. Moreover, nanomaterials are known to control infections in vitro and in vivo [30] via various mechanisms. Nanomaterials can act through the disruption of bacterial cell walls and membranes and damage specific membrane components such as efflux pumps. Nanomaterials can damage intracellular components such as DNA, and proteins through oxidative stress generated from ROS production. Moreover, nanoparticles (NPs) can inhibit the electron transport chain, prevent biofilm formation, and cause it to rupture once formed [31]. On top of those direct mechanisms, and of interest to this review, nanoparticles can be used to deliver drugs such as AMPs and synergize with them, increasing their safety and efficacy. The synergistic mechanism of action between nanoparticles and AMPs can be seen in Figure 1.

According to the present AMR scenario and considering the potential advantages of AMPs in combating bacterial infections, the present review aims to describe the therapeutic potential of AMP-based therapies for treating bacterial infections. It focuses on nanomaterial delivery systems for AMPs and highlights recent advances in AMP and nanoparticle conjugation approaches through different studies in this field in the last five years. We also discuss the current issues and possible solutions regarding AMP-based biomaterials.

## 2. Different Nanoencapsulation Methods for Efficient AMP Delivery

A variety of materials can be used to produce nanocarriers for AMPs, including inorganic (metal nanoparticles) and organic (polymer—and lipid-based) nanomaterials [30]. Figure 2 shows the main nanomaterials used to transport AMPs.

### 2.1. Inorganic Nanoparticles

Inorganic or metal nanoparticles have antimicrobial properties and, combined with AMPs, can improve their effectiveness against infections caused by various microorganisms, such as *Staphylococcus aureus*, *Pseudomonas aeruginosa*, *Candida albicans,* and *Escherichia coli* [30]. AMPs have functional groups (amino, carboxyl, and thiol) with high affinity for gold or silver atoms, which can immobilize peptides by electrostatic and hydrophobic interaction [32] and reduce the toxicity of the metal nanoparticle and, at the same time, increase the AMP activity [23]. Metal nanoparticles do not have a specific mechanism of action and do not bind to specific receptors on bacterial cells. This feature increases the spectrum of antibacterial activity and hinders the development of resistance by bacteria [32,33]. Among the various advantages that inorganic nanoparticles offer, we should mention the oxidative stress caused by ROS production that can act by the disruption of cell walls and damage of DNA and/or RNA molecules or even by inhibition of their synthesis [34].

Therefore, gold nanoparticles (Au-NPs) and silver nanoparticles (Ag-NPs) have been conjugated with AMPs improving the effects of both molecules, potentially reducing the toxicity of the metal nanoparticles and increasing the AMP’s efficacy [23].

Au-NPs exhibit many desirable features, such as their biocapacity and relative stability, which make them a good option for peptide nanoencapsulation [35]. In addition, Au-NPs have a small size, a large surface area, a high reactivity to living cells, good cell permeability, and antioxidant and anti-inflammatory properties in addition to their antimicrobial activity. They are widely used in wounds to reduce inflammation and promote healing and are versatile molecules for loading, transporting, and unloading various drugs in vivo, including peptides. They can exert catalytic effects to enhance antibacterial properties when combined with other antibacterial substances, such as AMPs [36].

Several studies have combined AMPs with Au-NPs to improve peptide stability and antimicrobial efficiency for treating bacterial infections. The peptide HuAL1, derived from complementarity-determining regions of monoclonal antibodies, was conjugated with Au-NPs. The conjugated HuAL1 showed the highest antimicrobial activity at peptide concentrations of 1 mg·mL^−1^ for *S. aureus* and 1.2 mg·mL^−1^ for *P. aeruginosa* [37]. The conjugation with Au-NPs showed antibacterial activity with lower concentrations than the peptide alone, which is very important since lower concentrations avoid side effects.

Another successful example involved the peptide Lys AB2 P3-His, a hexahistidine-tagged AMP successfully loaded onto DNA aptamer-functionalized gold nanoparticles (Au-NP-Apts) [38]. Conjugated Lys AB2 P3-His-Au-NP-Apt significantly reduced *Acinetobacter baumannii* colonization in murine organs, achieving a survival rate of 70% of infected mice. In contrast, in the control group treated with the peptide alone, the survival rate achieved was only 20%. In cytotoxicity assays, LysAB2 P3-His-Au-NP-Apt increased the number of viable HeLa cells by more than 3.6-fold compared to the control mice treated with the peptide or nanoparticle alone. The results led to a prominently increased survival time and rate in mice infected [38].

The peptide LL-37 was also successfully conjugated with Au-NPs. This AMP from polymorphonuclear leucocytes is a cathelicidin derived from the hCAP18 proprotein (inactive version), which, after processing, releases the active C-terminal sequence of 37 amino acids, the first two residues of which are leucine [39]. LL-37-Au-NPs inhibited the growth of *S. aureus* and caused an 85% wound-healing effect on day 12 of the trial compared to the control group [39].

Another peptide, the battacin analog Ura 56, was protected by adding a side chain PEG and successfully conjugated to Au-NPs [35]. Ura56-PEG-Au-NP conjugates were effective against methicillin-resistant *S. aureus* (MRSA), *E. coli*, multidrug-resistant *P. aeruginosa*, and *A. baumannii* at concentrations ranging from 0.13 to 1.25 μM. These concentrations were lower than the free Ura 56, demonstrating an increased potency in the peptide efficacy. This may be possible because the NP conjugation protects the peptide. They could also inhibit 90% of initial biofilms and 80% of preformed biofilms of *S. aureus* and *E. coli* at low micromolar concentrations. Furthermore, Ura56-PEG-Au-NPs demonstrated stability in rat serum while 45% of the free Ura 56 was degraded just 6 h after incubation, probably due to the difficulty of protease to bind to the Ura-56 attached to the surface of Au-NPs. Minor cytotoxicity in representative mammalian cell lines in vitro (≤64 μM) and in vivo (≤100 μM) was also achieved [35].

In addition to Au-NPs, AMPs combined with silver nanoparticles (Ag-NPs) have significant potential for treating bacterial infections. For thousands of years, silver has been used in the topical treatment of wounds and burns, and in recent times, it has gained increasing visibility due to its broad antibacterial action [33]. Ag-NPs are small and have a high surface area-to-volume ratio. Their initial contact with bacteria releases Ag^+^ and increases oxidative stress while having low toxicity against healthy cells [40].

The synthetic P-13 peptide was combined with Ag-NPs. The P-13-Ag-NPs showed effective antibacterial activity with minimal bactericidal concentration (MIC) values of 7.8 μg·mL^−1^ against *E. coli*, *S. aureus*, and *Bacillus pumilus* and 15.6 μg·mL^−1^ against *P. aeruginosa* while the MIC values for P-13 alone were much higher, implying a potent synergic effect on the bacterial activity. Moreover, Ag-NPs cytotoxicity was significantly reduced after being conjugated with the P-13 peptide, probably because the peptide covered the Ag-NP outer shell and reduced the metal surface contact with the cells. Moreover, P-13-Ag-NPs were more selective toward bacterial than mammalian cells [41].

Similarly, the synthetic Dpep peptide, originally designed from bactenecin, was also successfully conjugated with highly luminescent silver nanoclusters (Dpep-Ag NCs). Dpep-Ag NCs demonstrated enhanced inhibitory efficacy against *E. coli* at a MIC of 6.5 µM. In contrast, Ag-NPs alone (30 nm) worked at 800 µM and control nanoclusters complexed with bovine serum albumin (BSA-Ag NCs) were inhibitory at 100 µM, which indicates a synergistic effect between AMP and Ag. The conjugate allows better electrostatic interaction with the cell membrane of Gram-negative bacteria, generating better membrane permeability. The Dpep-Ag NCs accelerated wound closure in a murine model, with wound healing by 91% on day 5, demonstrating practical clinical application [42].

Another synthetic peptide, tryasine, was also conjugated with Ag-NPs to boost its antibacterial activity and demonstrated good antimicrobial and low hemolytic activity. Tryasine-Ag-NPs were more effective than tryasine alone against *S. aureus* and *E. coli* at 30 and 28 μg·mL^−1^ MICs, respectively. These values are around 50% lower than peptide alone, probably due to the peptide interaction with the membrane of bacteria, which causes the increase in permeability, leading to the antibacterial effect of Ag-NP. Tryasine-Ag-NPs exhibited 1% hemolytic action on human erythrocytes. Therefore, tryasine conjugation with Ag-NPs is a promising candidate for bacterial infection with low toxicity [43].

The designed (LLRR)3 cationic amphiphilic α-helical peptide has an antibacterial effect but a high hemolytic activity. Thus, (LLRR)3 was conjugated with Ag-NPs to overcome the latter [44] (Li et al., 2024). (LLRR)3-Ag-NPs had stronger activity against *E. coli* and *S. aureus* than the free AMP and unconjugated Ag-NPs, displaying MICs of 2.5 μg·mL^−1^ and 1.25 μg·mL^−1^, respectively. In contrast, the MICs of AMP and Ag-NPs against *E. coli* and *S. aureus* were 4 μg·mL^−1^ and 20 μg·mL^−1^, respectively. Interestingly, (LLRR)3-Ag-NPs reduced hemolysis, which occurred only at 12.5 μg·mL^−1^. The (LLRR)3-Ag-NPs also had lower cytotoxicity, with 80% cell survival in the tested conditions. Probably the AMP was responsible for the cytotoxicity reduction of Ag-NPs, covering the Ag-NP’s shell and reducing their contact with the cells [44].

Despite the advantages mentioned here, some drawbacks related to metal nanoparticles need to be overcome. The therapeutic effects in mouse models, such as the reduction in bacterial load and inflammatory damage in organs, are not usually as satisfactory as in vitro assays [32,33]. Another issue of metal NPs is their unpredictable toxic effects that threaten human health. The cytotoxicity of metal NPs is directly associated with their properties, such as size, shape, composition, charge, solubility, and coating material. Metal NPs can penetrate cells and interact with other molecules, such as proteins, causing neurotoxicity, immunotoxicity, and genotoxicity. Oxidative stress is one of the main mechanisms of cytotoxicity of metal NPs, caused by the excessive production of reactive oxygen species (ROS), which alters the oxidation-reduction state. Another mechanism is inflammation, a natural protective response to infection that can have detrimental effects if not regulated. Both mechanisms are related. Therefore, the biosafety of these materials is a major issue that needs to be solved for their wide clinical application [45].

The development of bacterial resistance is a natural process of adaptation and survival of bacteria. Bacteria have already demonstrated resistance against Ag-NPs through the secretion of the protein flagellin, which can induce the coagulation process of Ag-NPs and drastically reduce their antibacterial activity [46]. Strategies have been employed to overcome this challenge, such as binding silver nanoparticles to cyanographene. This combination was able to eliminate Ag-NP-resistant bacteria at a MIC value of 3.4 mg·L^−1^ against Ag-resistant *E. coli* compared to 108 mg·L^−1^ for Ag-NPs alone, 1.9 mg·L^−1^ against Ag-resistant *P. aeruginosa* compared to 54 mg·L^−1^ for Ag-NPs alone. The strong interaction between cyanographene and silver profoundly suppressed silver leaching [46]. Bacterial resistance against Au-NPs has also already been described, and strategies to overcome this issue have been proposed [47].

NPs from derivatives of metalloids like silicon [48] are also noted for their benefits in bacterial infection treatment [32]. Given their high surface area, robust framework, and porosity, mesoporous silica nanoparticles (MSNs), subtypes of Si-NPs, are considered excellent candidates for drug delivery. These result in a high density of internal pores, which can accommodate many antimicrobials [31]. MSNs have advantageous properties such as increased stability and easier preparation and modification relative to other nanocarriers. Their tolerance of changes in morphology and load, a wide variety of geometries, and range of porosity, biocompatibility, and biodegradability are also of great interest for therapeutic use [49]. One of the significant advantages of this system is that the well-defined pore structure allows control of drug loading and release kinetics, which prevents the degradation of AMPs by proteases [32].

The synthetic NZW peptide, which shows activity against clinical and multi-drug-resistant strains of *Mycobacterium tuberculosis*, was nanoformulated with mesoporous silica particles (MSPs) to improve its effectiveness against tuberculosis [50]. The results demonstrated that NZW-MSPs increased the inhibition of bacterial cells in infected primary macrophages compared to free NZW peptide and maintained their effectiveness in an in vivo murine model infected in lungs with *M. tuberculosis,* achieving a reduction of 88% [50].

Despite having many advantages for delivering AMPs, nanoparticles also present problems such as being phagocytosed or accumulating in the liver, which reduces their therapeutic efficacy and increases their toxic effect. Among the strategies that can be used to solve these problems, hybrid nanoparticles stand out. The use of different nanomaterials together, such as using another nanomaterial with inert or hydrophilic molecules to modify the surface of a nanoparticle, can overcome these problems [51,52], such as described above about binding silver nanoparticles to cyanographene. There are several possibilities for hybrid nanoparticles, such as polymer-metal, graphene oxide–metal, graphene oxide–polymer, and lipid–polymer hybrid nanoparticles [53]. Some examples of hybrid nanoparticles are described later in the manuscript.

MSNs are ideal for transporting different types of drugs together. Nano drug-delivery systems that co-deliver and co-release multiple combinations of agents favor synergy between different drugs, such as the combination of AMPs and antibiotics to enhance activity against (multi-drug-resistant) MDR infections [54]. Hybrid nanoparticles improve biological features, increase treatment efficacy, and decrease toxicity and resistance [31].

A co-delivery platform composed of MSNs with the peptide melittin and the antibiotic ofloxacin showed strong antibiofilm capacity [54]. The melittin peptide is present in bee venom and is a membrane-active peptide that inserts itself into membranes, causing cell lysis [55]. The MSN coassemblies promoted the sustained release of melittin, almost 70% in the presence of *P. aeruginosa* and continuous heating at 60 °C. Furthermore, MSN coassemblies removed >97% of the biomass of *P. aeruginosa* biofilms. In contrast, the control biofilms and those treated by unconjugated MSNs, free melittin, or ofloxacin had dense bacterial clusters. The MSN coassemblies also showed antibacterial efficacy in in vivo implantation models by eradicating pathogenic biofilms from implants [54]. Biofilms are difficult to access with antimicrobial molecules because pathogenic cells are protected by thick and dense extracellular polymeric substances (EPSs), composed of polysaccharides, proteins, glycoproteins, lipids, surfactants, and nucleic acids. This feature makes biofilms commonly resistant to multiple drugs [54]. These data demonstrate how the combination and synergy between AMP, antibiotics, and nanoparticles is relevant in treating resistant infections.

In a recent study, a synergic co-delivery system based on porous silicon (PSi) coloaded with the peptide et-213, an oligopeptide with a terminal thiol group, and Ag-NPs were developed [40]. The PSiMPs-et-213–Ag-NPs system displayed the highest antibacterial activity compared to control groups, with a MIC value of 1.5 mg·mL^−1^ against *E. coli* and 2 mg·mL^−1^ against *S. aureus*, respectively, with low toxicity to the host. It exhibited the best wound-healing effect in wounds of rats, showing no edema or inflammation, the highest percentage of decreased wound area among all tested conditions, and scabs formed after just three days of therapy [40].

Another AMP, NapFab, initially isolated from bronchoalveolar lavage and optimized in silico, has robust activity against *M. tuberculosis* but low intracellular availability. Therefore, it was loaded onto MSNs as a carrier system [49]. The MSN-encapsulated NapFab reduced the growth of intracellular *M. tuberculosis* within macrophages by 80% [49]. The conjugation to MSNs provides better cellular uptake and antibacterial efficacy.

A novel type of delivery system based on mesoporous fusiform nanoparticles, termed a gut-targeted engineered particle vaccine (EPV), has been designed for the targeted delivery of two hybrid AMPs, F6P1 and F6P6, to the intestine to treat *Clostridium perfringens* infection. The EPV delivery system was designed to increase the specificity and bioavailability of the antibacterial, antibiofilm hybrid AMPs (HAMPs). HAMPs conjugated with EPV exhibited more significant antimicrobial activity against *C. perfringens* colonic infections compared to HAMPs alone according to MIC and antibiofilm assays [56].

These studies demonstrate how the conjugation of AMPs with metal NPs significantlycontributes to the treatment of infectious diseases. This formulation allows the distribution of AMP in vivo, reducing its cytotoxicity and protecting it against degradation by proteases. AMPs conjugated with metal NPs provide reduced toxicity, greater antibacterial activity, superior targeting, and bett stability than free AMPs.

Table 1 briefly describes some AMPs conjugated to metal nanoparticles, including information about target infection and synergistic activity achieved.

### 2.2. Polymeric Nanoparticles

Polymer-derived nanomaterials are also widely used in AMP nanoformulation. They are small particles and increase AMP solubility, protect them from protease degradation, and prevent rapid renal filtration, prolonging circulation in the blood [32]. Polymers also enable controlled and sustained release of AMPs, and site-specific accumulation, thereby reducing dosage [33,57]. AMPs can be adsorbed, dissolved, encapsulated, or even attached to the polymer, released mainly through diffusion [58]. The easy synthesis of polymers in large quantities and at much lower costs are very attractive features for nanoformulating AMPs. The polymers also have a broad spectrum of antibacterial action and a low propensity for the development of bacterial resistance. Furthermore, they present greater stability in biological and storage conditions and controlled release [53].

One of the first polymers used for this and still in wide use is polyethylene glycol (PEG). Different PEGylation strategies have been employed to increase the structural stability of AMPs and, consequently, enable their arrival at the site of action [32]. Various studies demonstrate that the nanoencapsulation of AMPs with PEG significantly improves pharmacological properties, stability, therapeutic index, and water solubility. PEG conjugation also promotes lower renal clearance [23]. In addition, it provides other benefits, such as improvements in their capacity to form micelles for specific applications [59].

As an example, the terminally deaminated isoform of the maximin H5 peptide (MH5C), previously isolated from *Bombina maxima*, was altered at the C-terminus by a cysteine addition (MH5C-Cys) and then conjugated with PEG polymer weight of 2 kDa and 5 kDa [60]. The antibacterial assays showed that MH5C-Cys-PEG (5 kDa) has antibacterial action against *P. aeruginosa* and *E. coli* biofilm as assessed by MIC (40 μM) and minimal bactericidal concentration (MBC; 300 μM), while MH5C-Cys-PEG (2 kDa) did not show inhibition or eradication of the biofilms [60]. The exact reason for this weight dependency is unclear. These data reinforce the importance of carefully studying nanoparticles and further design with target AMP structure analysis.

Another peptide with potent bactericidal activity, N6, is a derivative of arenicin-3 isolated from the lugworm. Despite its potent bactericidal activity, it is not resistant to enzymatic hydrolysis, which makes its distribution in vivo unfeasible. Therefore, linear PEG molecules of variable chain length were added to the N- or C-termini or to the Cys residue of N6 to improve its stability [61]. PEGylated N6 at the C-terminus (*n* = 2) had potent activity against *E. coli* and *Salmonella pullorum* and greater stability against trypsin than unconjugated N6. It also showed a more potent immunomodulatory effect compared to N6 alone, reducing the levels of tumor necrosis factor-α (TNF-α) to 31.21%, interleukin-6 (IL-6) to 65.62%, and IL-1β to 44.12% and increasing the level of IL-10 to 37.83%. PEGylated N6 also demonstrated a wider biodistribution and longer half-life in mice than N6, improving the survival rate of infected mice [61]. Interestingly, the antibacterial activity of N6 PEGylated with more PEG lengths was reduced, possibly due to PEGylation changes in the structural features of peptides.

Nevertheless, PEGylation with longer PEG spacer lengths enhanced the antibacterial properties of the peptide KR12. The KR12, a derivative of LL-37, was immobilized on an anti-biofouling copolymer film and conjugated with different PEG spacer lengths. The antimicrobial properties of these conjugates against *E. coli* and *S. aureus* biofilm formation were then evaluated. The results demonstrated that the film with a long PEG spacer and high density of KR12 peptide decreased bacterial adsorption of *E. coli* by 95% in 24 h, indicating that a longer spacer length and a higher density of AMP resulted in better antibacterial properties since the long PEG spacer facilitates the KR12 access to the bacterial cell membrane [62]. These results point to the relevance of the individuality of each system, considering the peptide.

It is important to emphasize that despite PEG improving the stability of protease and prolonging the action time of AMPs, this material is not biodegradable and can accumulate in human organs and tissues, causing immunogenicity and limiting its therapeutic use. Some reports have shown that PEG can reduce peptide efficacy. Therefore, treating bacterial infections with AMPs associated with PEG is still a challenge in vivo [23,32].

Another polymer widely used for AMP encapsulation is poly(lactic-*co*-glycolic) acid (PLGA), a biocompatible synthetic polymer. Among the properties PLGA nanoparticles offer for peptide delivery are controlled AMP release, protection against premature degradation, cytotoxicity reduction, surface functionalization to target the infection site, or co-delivery with other bioactive molecules. They can also improve wound healing and are biodegradable, minimizing concerns about bioaccumulation [63].

An example is the synthetic peptide HHC10, which has excellent potential application against several Gram-positive and -negative bacteria strains. It was nano-formulated with PLGA to decrease its toxicity and enhance its therapeutic potential [64]. Stability tests demonstrated no peptide degradation during processing and extended-release over 180 h. The HHC10-PLGA nanoparticles significantly reduced *E. coli* growth over 12 h at the lowest concentration of 5 mM compared to the negative control group and the pure peptide. At the same time, the nanoformulation was not cytotoxic against mouse macrophage cells (RAW264.6) at up to 20 µM [64]. These data indicate that PLGA particles can reduce AMP toxicity of AMPs, probably because PLGA precisely delivers AMP to its site of action, preventing its interaction with macrophage cells.

Similarly, the AMPs 1-21 and 1-21-1c, both derivatives of esculentin-1 from frog skin, when encapsulated in PLGA nanoparticles coated with polyvinyl alcohol (PVA), were able to inhibit the growth of *P. aeruginosa* by approximately 60% up to 72 h. Here, PVA is adsorbed on the surface of PLGA and reduces the adhesion of hydrophobic PLGA to airway mucus. In a murine model of acute pulmonary infection caused by *P. aeruginosa*, the 1-21-PLGA and 1-21-1c-PLGA conjugates displayed more potent antibacterial activity, approximately 17-fold and 4-fold, respectively, compared to control groups. The conjugates could cross the bacterial extracellular matrix at approximately 68% after 1 h, and the diffusion of the Esc peptide from NPs through the bacterial extracellular matrix was 69.4%. At te same twhime, for free Esc, it was only 32.7% [65]. In addition to extending and increasing the therapeutic effect against *P. aeruginosa* lung infections compared to AMPs in their free soluble form, the nanoformulation enabled airborne administration.

Another AMP, the circular peptide AS-48, was conjugated to biomimetic magnetic nanoparticles (BMNPs) and PLGA, resulting in a PLGA[AS-48-BMNPs] nanoassembly [66]. The PLGA was used to enhance BMNPs’ internalization, since they present biocompatibility and biodegradability properties. Experiments showed that 78% of PLGA[AS-48-BMNPs] were successfully internalized by THP-1 cells in 72 h. Furthermore, the release of AS-48 from the nanoassembly reached ~45% at 37 °C and 7.4 pH, slower than previous nano-formulations. The formulation promoted an 80% increase in THP-1 viability compared to control groups. The formulation also improved the uptake, reduced the cytotoxicity, and increased the activity of AS-48 in treating macrophages infected with *M. tuberculosis* [66].

Similarly, the peptide SAAP-148, designed from LL-37, was formulated with PLGA nanoparticles to improve its selectivity and bioavailability at sites of infection and reduce its cytotoxicity [67]. The results indicated that the selectivity index of SAAP-148-PLGA nanoparticles was 10-fold higher against *S. aureus* and 20-fold higher against *A. baumannii* compared to free SAAP-148 after 4 h. Furthermore, the nanoparticles were 24-fold less cytotoxic against skin fibroblasts after 24 h and 10-fold less hemolytic against human erythrocytes [67].

In addition, the peptide OH-CATH30, a natural linear cationic peptide from the King Cobra, was conjugated with PLGA to manage murine enteritis. According to the results, PLGA-OH-CATH30 microspheres can potentially reduce intestinal damage and modulate the intestinal microbiota in enteric infections induced by *E. coli* [68]. They significantly decreased weight loss and intestinal damage, reduced infection-induced spleen index enlargement, normalized leukocyte and neutrophil levels, suppressed inflammatory cytokine release (IL-1β, IL-6, and TNF-α), and raised the anti-inflammatory cytokine IL-10. In addition to reducing the abundance of pathogenic bacteria in the intestinal tract, such as *E. coli*, beneficial bacteria increased in treated animals [68].

Many other studies report the benefits of PLGA for the nanoformulation of AMPs. NZ2114, a plectasin derivative, was loaded onto a PLGA–PVA drug delivery system. PLGA encapsulation increased cell viability by 20%, NZ2114 retention by 50%, and trypsin resistance by 4.24-fold. Moreover, NZ2114-PLGA had inhibitory activity against *Staphylococcus epidermidis* at 4–8 μg·mL^−1^ and effectively reduced the biofilm and *S. epidermidis* population at a rate of 99% [69]. Other peptides have successfully been conjugated with PLGA and had their properties improved, such as OVTp12 [70] and OP-145 [71].

These studies demonstrate how the conjugation of AMPs with PLGA NPs has made significant contributions to the treatment of infectious diseases. It protects the AMP against degradation by proteases, provides long-lasting release of the target AMP at the site of infection, and reduces cytotoxicity.

Among the other polymers used for AMP nanoformulation, chitosan merits mention. It is a linear polymer consisting of N-acetyl-D-glucosamine and β-(1,4)-linked D-glucosamine [72]. It is a naturally occurring, biocompatible, biodegradable polysaccharide that promotes strong adhesion to the mucosa and anti-infective activity. In addition, chitosan itself has antimicrobial activity and low toxicity. Chitosan has a cationic structure due to positive amino groups, enabling interaction with dmagnegating compounds such as bacteria membranes [58]. It also protects AMPs from degradation during administration and sustains their release, thus increasing bioavailability [57,73]. Also, it has a low cost. Chitosan is specially used for wound healing, promoting wound closure rate, neovascularization, re-epithelialization, and cellular proliferation at the location of the wound [58]. Various authors have indicated that chitosan is a promising candidate for AMP encapsulation in treating infections [57].

Originally isolated from wasp venom, mastoparan was encapsulated with chitosan nanoparticles to enhance its stability and therapeutic efficacy against MDR *A. baumannii* clinical isolates [74]. The mastoparan-chitosan-NPs demonstrated biocompatibility even at higher concentrations; hemolysis was not observed. They retained 98% activity for up to 72 h, whereas free mastoparan was only 42–56% after that time. Mastoparan-chitosan-NPs were able to kill *A. baumannii* cells at 4 μg·mL^−1^ while free chitosan needed 512 μg·mL^−1^. A significant reduction in colony-forming units (CFUs) was observed in a sepsis model in BALB/c mice treated with mastoparan-chitosan-NPs compared to chitosan and mastoparan alone [74].

Another successful example is octominin, a synthetic AMP designed from a protein of *Octopus minor* and known for its action against *Candida albicans* and *A. baumannii* [75]. Octominin was also successfully nanoencapsulated with chitosan and carboxymethyl chitosan. Chitosan nanoencapsulation was used to overcome poor stability and toxicity. Octominin-chitosan nanoparticles preserved cellular viability of 97.83% at the highest tested concentration (400 μg·mL^−1^), while for free octominin it was 85.19%. According to time-kill kinetic assays, they had slightly higher action than octominin alone at six hours of treatment. Also, they caused a reduction to 8% in *C. albicans* viability, whereas free octominin caused a decrease to 11% [75].

Similarly, cecropin-B, previously isolated from *Hyalophora cecropia* and produced in *E. coli* using recombinant DNA technology, was encapsulated with chitosan particles to increase its lifetime and improve its targeting and efficacy against MDR *K. pneumoniae* [76]. The lowest MIC of cecropin-B-chitosan particles obtained among the isolates evaluated was 1.6 μg·mL^−1^, more effective than a free peptide. It is essential to highlight that the free capsule showed a mild cytotoxic effect on bacterial cells at the highest concentration, corroborating that chitosan has a antibacterial impact itself and can improve efficacy conjugated with an AMP. Also, the cecropin-B-chitosan particles demonstrated zero hemolysis at 6.25 μg·mL^−1^ [76].

Another example is the C7-3 AMP and its derivatives C7-3m1 and C7-3m2, which were successfully formulated in chitosan nanoparticles and have shown promising activity against MDR strains of *Neisseria gonorrhoeae* [72]. The assays indicate that chitosan NPs loaded with C7-3, C7-3m1, and C7-3m2 enhanced anti-gonococcal and anti-biofilm efficacy. Furthermore, the NPs demonstrated cytocompatibility in HeLa cell lines, with no cytotoxicity observed [72]. Hydrogels are also used for AMP delivery. These biomaterials have different medical/biomedical applications, mainly as dressings, but they also possess antimicrobial efficacy, preventing colonization in the wound [59]. Hydrogels do not need toxic organic solvents in preparation, avoiding residual material, and have controlled and responsive drug release [58,77]. Hydrogels are polymer networks swell with water, shaped by crosslinking hydrophilic polymer chains in an aqueous microenvironment [77]. The polymers used here can be natural, such as hyaluronic acid and alginate, or synthetic, such as poly (ethyl acrylate-co-methacrylic acid) and poly (N-isopropylacrylamide) [58]. They have a similar structure to the extracellular matrix, can absorb wound secretions, and contribute to ambient humidity around the wound, regulating the wound microenvironment and thus promoting skin healing. Another significant advantage of hydrogels is that they can, at the same time, be used as carriers of antibacterial compounds such as AMPs to improve the skin healing rate [78]. Their porous structure delivers AMPs at specific locations for slow-release purposes [32]. Therefore, peptide hydrogels are potential biomaterials with broad application in medicine, specifically against infectious diseases, due to their excellent biocompatibility, injectability, and deference to 3D printing.

HHC36, a short-designed peptide, was loaded into a macroporous composite hydrogel to evaluate its potential to deliver AMPs for effective bacterial inhibition [79]. The hydrogel showed a slower release of HHC36 than controls, which is excellent for long-term antimicrobial activity. Also, HHC36 released from the hydrogel killed almost 100% of *S. aureus* cells after six days of treatment, and no bacterial colonies were observed on the corresponding agar plates. After being treated with the HHC36-loaded hydrogel for 12 days, mice with dorsal skin wounds had almost completely recovered, showing a better healing performance than control groups [79].

The peptide ε-polylysine, known for its versatile antibacterial properties, create a nanocomposite hydrogel. This hydrogel, prepared with oxidized alginic acid and dopamine and cross-linked with acrylamide, was designed to combat bacterial infections. The ε-polylysine peptide enabled the hydrogel to effectively inhibit the growth of *E. coli*, *S. aureus*, and *P. aeruginosa* at various concentrations [80]

Jelleine-1 is a D-enantiomer of a parental peptide initially isolated from the royal jelly of a honeybee, that was also nanoformulated with a hydrogel. The jelleine-1 peptide hydrogel exhibited potent in vitro antimicrobial activity against *E. coli* and *S. aureus*. In an animal model, infected tissues treated with it yielded fewer bacterial colonies than the control groups. It also showed excellent blood clotting properties in a mouse model of hepatic hemorrhage, stopping bleeding significantly faster than gauze or no treatment. Additionally, it showed anti-adhesion efficiency and good biocompatibility [81].

Moreover, synthetic AMPs called V-Os were combined with collagen, a hydrogel dressing of methacrylate gelatin (GelMA), and the conductor of electricity Ti3C2 to improve wound healing caused by *S. aureus* and *E. coli.* The results demonstrated that a hydrogel dressing (GelMA@Ti3C2/V-Os) with a total peptide concentration of 62.5 µg·mL^−1^ provides a 50% antibacterial clearance rate. The hydrogel dressing (GelMA@Ti3C2/V-Os) significantly increased the expression of genes related to fibroblast migration, proliferation, and tissue repair [78]

The bioactive peptides Tet213 and QK were conjugated in a hydrogel tissue sealant AN@CD-PEG&TQ, which consists of four-arm PEG-succinimidyl carbonate (PEG-SC) and the AN@CD nanoprobe [82]. The resulting conjugated hydrogel system was evaluated against *E. coli* and *S. aureus* infections and caused a significant CFU reduction and a substantial inhibition zone compared to control groups. Furthermore, the hydrogel-based tissue sealant demonstrated activity in mouse models of liver hemorrhage, gastric perforation, and bacterial infection of skin wounds, showcasing its potential as a high-performance wound sealant for treating bleeding organ wounds [82]. A recent advancement in hydrogels is the redox-degradable hydrogel used for polymer-based hydrogels. Disulfide bonds are introduced into a polymer structure. They can be cleaved in response to variations in the environmental redox state, such as reactive oxygen species (ROS) produced in the wound, inflammation, or sites of bacterial infection and biofilms. Thus, the release of the therapeutic agent and degradation of the hydrogel is stimulated. The AMP vancomycin, a glycopeptide antibiotic medication, was conjugated in a redox-degradable hydrogel to treat skin infections topically and overcome the challenges of high doses needed for intravenous administration. The redox-degradable hydrogel loaded with vancomycin showed an effective, long-lasting antibacterial activity against *E. coli.* In vivo, the wound healing model assay showed that the percentage of wound contraction after 3 days of surgery was 58.65 ± 15.1 for 8% hydrogel + vancomycin, while for the control group, it was 36.3 ± 18%, indicating the importance of redox-degradable hydrogel [83].

An unnamed peptide was also nanoformulated with a type of precisely controlled-release hydrogel in response to environmental factors such as reactive oxygen species (ROS) and matrix metalloproteinases (MMPs) infection. The AMP was conjugated with a hydrogel composed of hyaluronic acid modified with cyclodextrin (HA−CD) and adamantane (Ad−HA). Ad-HA-AMP improved AMP stability and antimicrobial activity against *E. coli* and *S. aureus* with a live bacteria rate of less than 20%. The controlled release of AMPs induced by the MMPs and ROS promotes cell viability of more than 98%, while the control group caused cytotoxicity, decreasing cell viability by less than 80% due to uncontrolled release. Furthermore, the diabetic chronic wound model in in vivo assays demonstrated that wound healing was improved as assessed by wound diameter (less than 35%) and the presence of bacteria (less than 10%) [84].

Hydrogels are classified as macroscopic hydrogels, nanogels, and microgels according to their size [58]. Nanogels are promising carriers due to their excellent drug-loading capacity, greater stability, and ability to reach a specific site. This delivery system is a type of porous hydrogel and can be composed of different polymers. The nanogel particle size is adjustable since they are hydrated particles that can shrink and swell under different external stimuli, contributing to an effective controlled drug release. They are used in vadifferious therapeutic and diagnostic applications [85]. Due to existing Food and Drug Administration (FDA) approval, hyaluronic acid-based gels have become very attractive for rapid clinical application [86].

A novel AMP, designed and chemically synthesized, was successfully nanoencapsulated in a hyaluronic acid-based nanogel with nitric oxide (NO) for co-delivery against bacteria and biofilms [86]. According to in vitro antimicrobial assays, the resulting conjugated nanogel had MIC values of 1.56, 0.78, and 0.39 μg·mL^−1^ against *E. coli*, MRSA, and *P. aeruginosa*. Furthermore, it reduced MRSA biofilms 12.5-fold and *P. aeruginosa* biofilms 24-fold in catheters relative to NO alone. The nanogel loaded with only NO reduced MRSA biofilms 7-fold and *P. aeruginosa* biofilms 9.4-fold, also relative to only NO. The data showed that the nanogel loaded with NO and the novel AMP has excellent potential to combat bacterial infections and biofilms caused by resistant bacteria [86].

Several studies demonstrate AMP–hydrogel and –nanogel formulations. SAAP-148, a synthetic AMP, and Ab-Cath, a snake cathelicidin, were encapsulated in oleyl-modified hyaluronic acid (OL-HA) nanogels. Although the resulting NPs exhibited in vitro activity similar to that of free SAAP-148 and Ab-Cath solutions against AMR *S. aureus* and *A. baumannii*, there was a decrease in cytotoxicity, thus improving SAAP-148 selectivity 2-fold and Ab-Cath by 16.8-fold. The selectivity of Ab-Cath-loaded OL-HA nanogels reached 300 or more for *S. aureus* and 3000 or more for *A. baumannii* [87]. Similar studies have demonstrated nanoencapsulation of these peptides with other variations of hyaluronic acid nanogels. SAAP-148 was successfully encapsulated in hyaluronic acid nanogels modified with octenyl succinic anhydride [88], while Ab-Cath was successfully encapsulated in nanogels based on hyaluronic acid alone [89].

A class of nanoparticles based on star-shaped peptide polymers consisting of lysine and valine residues is also used to treat bacterial infections. Termed structurally nanoengineered antimicrobial peptide polymers (SNAPPs) are highly stable compared to other polymers. SNAPPs in the form of 16- and 32-arm star peptide polymer nanoparticles showed a broad spectrum of activity against Gram-negative bacteria such as *E. coli*, *P. aeruginosa*, *K. pneumoniae*, and *A. baumannii* with MBC values lower than 1.61 µM while demonstrating no significant cytotoxicity. The SNAPP 16 showed more than 99% of bacterial cell eradication in a mouse peritonitis model infected with *A. baumannii,* whereas only 20% of the control survived after 24 h. SNAPP 16 improved host cell innate immunity to *A. baumannii* in vivo by enhancing neutrophil infiltrate in the peritoneal cavity, while the control group shows no significant difference. According to the analyses performed, no microbial resistance was observed by colistin-resistant and MDR (CMDR) pathogens to the SNAPP since these nanoparticles presented different mechanisms of antibacterial action [90].

SNAPPs are also prospected to be applied for pulmonary delivery against respiratory bacterial infections such as pneumonia and tuberculosis. SNAPPs immobilized by different techniques in polyphenol-based capsules were internalized by alveolar macrophages in vitro. They were effective against *E. coli* with MIC values of approximately 30 μg·mL^−1^ with sustained release and non-significant cytotoxicity. Furthermore, they remain stable in nebulized droplets [91].

Another polymer used to deliver AMPs is poly(lactic acid) (PLA). This polymer has very favorable characteristics as a delivery system including biodegradability, compatibility with biomolecules and cells, and low production cost. Given this, PLA-based micro- and nanofibers have been used for wound healing [92]. The AMP temporin L isolated from the skin of the frog *Rana temporia* has potent antibacterial activity with MIC values ranging from 0.3 to 3.6 µM for various bacterial strains [93]. Therefore, temporin L was conjugated to a cationic-based polymer for potential application in wound dressing. Initially, the temporin L peptide was functionalized with a polymer containing talcin and thiazolium groups, forming the peptide conjugate polymer (PTTIQ-AMP). This conjugate was subsequently incorporated into PLA electrospun fibers to analyze the synergic activity. The PTTIQ-AMP conjugated into PLA fibers has shown improved antibacterial performance been capable of reducing *E. coli* and *E. faecalis* cells to 99.999% compared to control groups. This result can be attributed to diffusion capacity and leaching proprieties contributing to the effective and sustainable release of temporin L [93].

Table 2 briefly describes some AMPs conjugated to polymeric nanoparticles, including information about target infection and synergistic activity achieved.

Despite the considerable strengths of polymeric nanoparticles, they present some drawbacks such as the toxic organic preparation and degradation, and the residual material causing immunogenicity [58].

### 2.3. Lipid Nanoparticles

Another option for nanoformulating AMPs is lipid nanoparticles. These are small spherical vesicles with high surface area and are composed of ionizable lipids. Lipid nanoparticles usually have high biocompatibility, bioavailability, biodegradability, solubility, and reduced toxicity. These nanoparticles include liposomes, micelles, and liquid crystalline nanoparticles [58,63]. In addition to nanoparticles composed only of solid lipids, there are nanostructured lipid carriers, which are composed of a mixture of solid and liquid lipids that improve the solubility of lipophilic compounds. Both kinds are biodegradable, non-toxic, and versatile, carrying a range of chemically different bioactive substances, including peptides and proteins [59,94,95].

Liposomes comprise vesicles of amphiphilic phospholipids and cholesterol. Among the advantages of this delivery system is that its properties can be adjusted through lipid composition or the coating of the liposome surface. The dual polarity of liposomes due to the hydrophilic lipid head and hydrophobic tail allows the incorporation of hydrophilic and hydrophobic molecules [33]. Liposomes show high encapsulation efficiency, enhanced release, and antimicrobial effect [34]. Liposomes are biocompatible and biodegradable nanocarriers that are being applied for topical, oral, pulmonary, and systemic delivery [58].

In addition, since the delivery of AMPs by liposomes is mainly through adsorption or endocytosis, it provides greater penetration into tissues to combat intracellular infections. Liposomes reduce toxicity and improve the tissue uptake of AMPs, improving their biodistribution in vivo [96,97]. Therefore, encapsulating AMPs in liposomes is an attractive approach to prevent the disadvantages associated with the direct application of these molecules alone [77].

The peptide microcin J25 has a bactericidal action against pathogenic enteric bacteria such as *E. coli* and *Salmonella*. It was encapsulated in negatively and positively charged liposome models, double-coated with the biopolymer pectin and whey proteins to improve their stability and enable gradual delivery. Liposomal formulations protected the peptide during simulated gastrointestinal digestion. The amount of microcin degraded after four hours was below 50%, less than that of free microcin [97].

Similarly, the peptide thuricin CD, produced by *Bacillus thuringiensis*, was encapsulated in anionic liposomes to increase its solubility in the intestinal fluid. The peptide-loaded liposomes showed increased activity compared to the free peptide and blank liposomes, ultimately killing the bacterium *Listeria monocytogenes* at a concentration of 2.5 μg·mL^−1^. Furthermore, the peptide was not degraded when exposed to pepsin in gastric and intestinal fluid and was stable in suspension for more than 21 days. The data indicate that thuricin CD-loaded liposomes are a promising approach for oral administration [98].

The advantages of liposomes for the delivery of AMPs are undeniable. However, these vectors still run the risk of phagocytosis and clearance by the reticuloendothelial system. Strategies such as surface functionalization to induce charge inversion or size control can be used to overcome phagocytosis [25]. AMPs loaded in liposomes can improve their therapeutic efficacy by modifying the surface liposome, covering it with PEG, introducing other moieties, or coating it with antibacterial agents [96]. PEGylation of the surface of liposomes prolongs circulation time and reduces their uptake by macrophages [58]. Alyteserin-1c (called a +2 peptide), isolated from skin secretions of the frog *Alytes obstetricans*, demonstrated antibacterial activity against *E. coli*, with a MIC of 25 μM. Due to its physicochemical features, alyteserin-1c was used as a model for the design of a +5 cationic peptide version. To improve the stability of these peptides for their use in food preservation, they were encapsulated in liposomes coated with the Eudragit polymer. The results demonstrated that the antibacterial activity of the +2 and +5 peptides loaded in liposomes exhibited reduced MIC values against *E. coli*, of 1.25 and 5 μM, respectively. These data indicate that liposome vehicles prevent peptide degradation and favor their release near the bacterial surface [99].

However, these strategies can negatively impact the activity of liposomes, decreasing the efficiency of immune cell recruitment and impairing the reduction in inflammation or the elimination of bacteria [25].

Micelles are lipid-based carriers that also show promise for AMP delivery. They are structured as self-assembled spheres of single-layer lipids of surfactants. Their amphiphilic properties allow a high load capacity and better distribution in vivo. However, the incorporation of hydrophobic AMPs is limited due to the structure of the micelles [58]. PEGylated phospholipids are most used for this purpose, as they form spontaneously stabilized micelles with a hydrophobic 1,2-Distearoyl-sn-glycero-3-phosphorylethanolamine (DSPE) nucleus surrounded by hydrophilic PEG molecules. Due to increased solubility and specificity, delivery systems can improve the effectiveness and reduce the cytotoxicity of AMPs, enhancing their bioavailability [63,96].

The novel peptide DP7-C, derived from the highly active AMP DP7, was modified by adding cholesterol, forming an amphiphilic compound. DP7-C self-assembles into stable nanomicelles in an aqueous solution. DP7-C micelles had minor hemolytic activity in mouse assays, being tolerated at a concentration of 80 mg·kg^−1^ of body mass after 144 h by intravenous administration, while just 20 mg·kg^−1^ of unconjugated DP7 was enough to kill the control group within 10 min. Furthermore, DP7-C micelles demonstrated immunomodulatory activities in infection models in zebrafish and mice against *P. aeruginosa* and MRSA [100].

Another modified peptide, peptide 73, is derived from aurein 2.2. Different variations of peptide 73 were generated and formulated with PEG-modified phospholipid micelles. The micelle formulations showed reduced aggregation and toxicity against human cells. They were well absorbed when injected under the skin of mice, whereas in control groups, the non-micelle-encapsulated peptides clumped. Peptide 73 formulated in micelles reduced abscess size by 36% and bacterial loads by 2.2-fold compared with aurein 2.2 in a murine model of skin abscess by *S. aureus*. The modified peptides, 73c and D-73, reduced abscesses by 85% and 63% and decreased bacterial loads by 510-fold and 9-fold, respectively, relative to peptide 73 alone [101].

Liquid nanostructured crystalline particles can also be AMP carriers [77]. Liquid crystalline nanoparticles (LCNPs) are lipid bilayers that bend to acquire two-dimensional and three-dimensional structures with interwoven water channels. The biodegradable lipid glyceryl monooleate (GMO) is widely used for the manufacture of LCNPs. Examples of LNCPs are cubosomes and hexosomes, and several AMPs have been successfully loaded on them [77]. This system has advantageous characteristics such as greater solubility, bioavailability, and stability [58]. The use of cubosomes as a delivery vehicle was investigated for gramicidin A, alamethicin, melittin, pexiganan, cecropin A, and indolicidin peptides. Significant antibacterial activity was observed with phytantriol cubosomes loaded with indolicidin, achieving a MIC of 8 mg·mL^−1^ against *S. aureus* and 4 mg·mL^−1^ against *Bacillus cereus*; this represents a decrease in MIC of at least twofold relative to unencapsulated peptides [102].

Lipidic nanoparticles face other drawbacks such as polymeric changes and premature AMP release [58]. A problem that lipidic nanoparticles face in general, as do other delivery vehicles, is endosome escape. Delivery vehicles can enter early endosomes and be either sent back to the plasma membrane or proceed to the lysosomal pathway and thus cause degradation of the delivery system. Extracellular vesicles (EVs) may be an alternative to this problem. EVs are membrane-bound particles naturally secreted by various cells and are subclassified as exosomes, microvesicles/ectosomes, and apoptotic bodies. They can package and transport various bioactive molecules, making them promising molecules as delivery systems. Among the molecules transported by EVs are lipids, nucleic acids, and proteins [103]. In addition to AMPs being encapsulated by EVs, they can also be delivered by surface modification coating the EVs. Specific EVs were coated with a novel cationic AMP, AMP-A, that has good antibacterial and biocompatibility properties. This change made the surface charge of the vesicles neutral. This physical change improved the antibacterial activity against *E. coli* showing MBC values 2-fold lower in comparison to AMP-A alone. It also improved its biocompatibility and reduced the peptide’s cytotoxic effect [104]. The data indicate EVs as a potential alternative to improve the antibacterial activity and cytocompatibility of AMPs. Another EV type, a rose-derived exosome-like nanoparticle, was used to encapsulate ELNs AMPs. The nanoconjugates promoted enhanced antibacterial activity against intracellular MRSA, 2.5 times greater than ELNs alone, in in vitro cell infection assays [105].

### 2.4. Other Types of Nanomaterials

Dendrimer, cyclodextrin, and aptamer conjugates have also been successfully investigated to deliver AMPs [59,77]. Dendrimers, as well as their reduced forms (dendrons), have been explored as nanocarriers because they have a low manufacturing cost, and their synthesis is relatively simple. They have high structural precision with multiple terminal groups, which can be modified to modulate their physical, chemical, or biological properties. Given the chemical structure, the most common are polyamidoamine (PAMAM), polypropylene imine (PPI), poly-l-lysine (PLL), polyglycerol (PG), poly(benzyl ether), and carbosilane or phosphorus dendrimers [59,77]. Three different AMPs were fused with first- and second-generation cationic carbosilane dendrons with a maleimide molecule at their focal point [106]. The results suggest a synergistic effect on antibacterial activity against *S. aureus* and *E. coli* when the second-generation dendron is conjugated to an AMP [106].

Many authors have reported the use of cyclodextrins for the delivery of AMPs in recent years [77]. These are cyclic oligosaccharides composed of several dextrose units linked by α-1,4-glucosidic bonds in a hollow structure, and they are hydrophobic on the inside and hydrophilic on the outside. This structure provides biocompatibility, solubility, and stability [77]. Alamethicin encapsulation in γ-cyclodextrin promoted excellent solubility plus temperature and pH stability relative to the AMP alone, as well as good antimicrobial activity against *L. monocytogenes* [107].

Also known as chemical antibodies, nucleic acid aptamers are single-stranded RNA or DNA molecules composed of 20–80 nucleotides. They can bind to selected target molecules with high affinity because of their three-dimensional structure [78]. These molecules are seen as promising components for biosensors or for tagging molecules for treatment, imaging, and AMP distribution. This is due to some of their advantages, such as being produced with no animal-based steps, their small size, and ease of modification [108,109]. Au-NPs conjugated with a histidine-tagged DNA aptamer have been shown to eliminate intracellular *Salmonella Typhimurium* in HeLa cells, increasing cell viability [110].

An antibody targeting *E. coli* was used to develop antibody–bactericidal macrocyclic peptide conjugates (ABCs), using the AMPs CAP-18, SMAP-29, and BMAP-27, from the cathelicidin family. The ABCs were effective against *E. coli* at nanomolar concentrations and had minimized hemolytic activity [111].

The strengths and weaknesses of the main nanoparticles used for AMP delivery are described in Table 3. Composition and route of administration are also included.

The choice of nanomaterial is critical since nanoparticles interact directly with the plasma membrane of cells and this is reflected in the uptake of nanoparticles. The plasma membrane comprises a phospholipid bilayer containing several biomolecules and is mostly negatively charged. Given this, it has selective permeability to ions, biomolecules, and nanoparticles [112]. Therefore, it is essential to take into account the chemical, physical, and structural characteristics of the nanomaterial, as well as the entry mechanism, since all of this determines the entry and performance of the function of the nanoparticle in question. Nanoparticles cross the plasma membrane by endocytosis-based absorption pathways and direct cellular entry. Endocytosis consists of different pathways and mechanisms mediated by lipids and transport proteins, enabling nanoparticles to cross cell membranes and enter cells. After being endocytosed, nanoparticles are stored in endosomes, phagosomes, or macropinosomes and do not have direct and rapid access to the cytoplasm or cell organelles. Direct access through delivery channels allows nanoparticles to access the entire cytoplasm and interact with intracellular organelles and structures, facilitating the performance of their specific biological functions [112].

## 3. Synergistic Effect of Nanoformulated Peptides

Silver and zinc oxide nanoparticles release metal ions, which can act through electrostatic interactions with bacterial membranes, destabilizing the intracellular redox balance and causing DNA damage [113]. Similarly, AMPs have activity against a broad spectrum of microorganisms and are essential for the innate and acquired immune system, functioning as a defense mechanism. Because most active AMPs are cationic, they also act on the bacterial plasma membrane, including that of multidrug-resistant (MDR) bacterial strains, leading to membrane depolarization and cell permeabilization [28]. However, both NPs and AMPs have limitations that need to be overcome. Silver nanoparticles, for example, in addition to having low stability in aqueous systems, are prone to aggregation and can be cytotoxic to healthy cells [44]. Otherwise, AMPs may have underlain systemic toxicity [113]. Therefore, new strategies to overcome these limitations need to be developed.

In this scenario, the development of nanoparticle-based systems, which enable the controlled release of drugs or natural substances (such as peptides), is a promising strategy to improve the antimicrobial activity of these compounds [114]. Biomolecules such as proteins, polysaccharides, and peptides with –OH, –COOH, and –NH_2_ functional groups can bind to metal nanoparticles through intermolecular hydrogen bonds, adsorbing onto their surfaces. This interaction increases silver nanoparticle stability in solution [44]. Another example is mesoporous silica nanoparticles (MSNs), which have advantages as drug carriers such as high porosity, low toxicity, and relative ease of chemical modification on their surface. These characteristics allow the development of complex encapsulated systems that can deliver the cargo on command by applying a specific external stimulus [114].

Recently, there has been an increase in publications demonstrating the improvement of the nanoformulation peptides’ antimicrobial activity due to synergism [115,116,117]. It was observed that chitosan/bioactive glass nanoparticles/tetracycline composite coatings coated on the etched substrate and with the subsequent addition of melittin by dripping eradicated adherent bacteria and prevented biofilm formation on the implant surface. This fact demonstrates that there is a synergism between the melittin peptide and tetracycline when in a chitosan/bioactive glass coating nanoformulation, which can be explored as a multifunctional coating for bone implants [118]. Furthermore, the synergistic effect can be exploited for treating burn wound infections as demonstrated by Wali et al. in a study in which a combined dressing of decellularized human amniotic membrane (dHAM) loaded with colistin and Ag-NPs was developed. The results indicated that dHAM associated with colistin and Ag-NPs had a better performance in combating *P. aeruginosa* and *K. pneumoniae* when compared to the control in infected rats [119].

The combination of peptide, nanoparticle, and a conventionally used antibiotic can also be explored as an anti-biofilm. The IDR1018 peptide, when nanoformulated with chitosan nanoparticles (CNs) and in combination with ciprofloxacin, showed an increased antibacterial and antibiofilm potential against clinical isolates of uropathogenic *Escherichia coli* (UPEC) resistant to ciprofloxacin [120]. Another approach is to exploit the synergism of peptides and nanoparticles to perform dual functions, as demonstrated by Wu et al. They developed zinc oxide (ZnO) nanoparticles with the glucagon-like peptide-1 (GLP-1) analog liraglutide (LG). ZnO is known to exert antibacterial activity against several strains, while LG acts to promote vascularization and wound healing. ZnO nanoparticles (ZnO-NPs) and LG were shown to simultaneously induce antibacterial, hemostatic, and vascularization effects for the healing of infected wounds [121].

Furthermore, the association between peptides and nanoparticles can decrease AMPs’ cytotoxicity against healthy mammalian cells without affecting their antibacterial activity. Chitosan derivatives are capable of attaching different dendrimers of the G3K peptide and were efficient in eradicating *P. aeruginosa* cells for more than 24 h, in addition to reducing the cytotoxic effect, demonstrating synergism [122]. Another approach that can be taken is using the peptide as a matrix material for the tips of the microneedles, functioning as an excipient for drug delivery [117]. Recently, the peptide ε-poly-L-lysine (EPL) was used to produce dissolving microneedles that facilitate the intracellular accumulation of the antibiotic doxycycline (DOX) by increasing the permeability of the bacterial cell membrane. Using models of deep culturing infection induced by *Pseudomonas aeruginosa*, the EPL microneedles exhibited synergism, increasing the antimicrobial activity and prolonging the retention of DOX in the infected lesions. A 99.91% reduction in bacterial load was observed in a single administration [116]. All the examples listed above show nanoformulated peptides’ potential to treat different diseases. However, some limitations make translating nanoformulated peptides into the clinic difficult.

## 4. Clinical Translation of Nanoformulated Peptides

Liposomal doxorubicin (Doxil^®^), used for cancer treatment, was the first nanoparticle-based drug launched on the USA market in 1995 [123]. Since then, more than 30 nanoformulations have already been approved, and more than 100 clinical trials are underway for the application of nanoparticle-based therapies, demonstrating the enormous therapeutic potential of NP-based therapies [124]. Among the nanoparticle-based drugs approved by various agencies in both Europe and the USA, there are several to treat different types of cancer, as well as iron deficiency anemia, ultrasound contrast agents, and, more recently, the development of two vaccines for COVID-19 based on lipid nanoparticles [125].

With their promising potential, peptides offer a beacon of hope in overcoming inherent limitations that can impede the clinical translation of antimicrobial agents, particularly in the context of antisepsis. Their ability to enhance stability and cellular internalization [104] is a significant stride forward. The unique physicochemical characteristics of peptides enable precise targeting, thereby reducing off-target effects. When combined with nanoparticles, these conjugates can be designed to be biocompatible and stable, thereby improving their pharmacokinetics and paving the way for clinical translation [126].

However, despite the increasing number of annual publications on nanoparticle drug-delivery systems, there are still many challenges to translating nanomedicine into clinical practice [127]. Limitations in the translation of NPs for clinical use occur due to the need for nanopharmaceuticals design, which must necessarily consider the NP’s physical and chemical stability, its biodegradability, sophisticated formulation, and route of administration; in other words, it is necessary to know the in vitro and in vivo effects, biodistribution, pharmacokinetics, and pharmacodynamics of NPs [128,129]. Large-scale production of NPs is another obstacle in the clinical translation of nanomedicine, as reproducibility is needed without a high production cost. In addition, polydispersity, scale-up complexity, incomplete contaminants purification, consistency and storage stability of the final product, morphology, and charge are other obstacles that need to be overcome [129]. In addition, there are other short-term challenges, such as: (i) our current understanding of the pathogenesis of infectious diseases is not sufficiently deep, making it difficult to act in rapidly changing conditions; (ii) the vast gap between animal models and humans; and (iii) the need for improvements in current technology for clinical design and translation [130]. They are associated with all these issues; properties of NPs such as their size, shape, net surface charge, peptide characteristics, and conjugation chemistry can significantly limit their specificity and accuracy in theranostics [126].

Safety and regulatory compliance implications and the unique properties and potential risks of NPs limit their application in clinical use, as extensive testing and evaluation are mandatory before nanoparticle-based drugs are approved for clinical use [131]. Although there are many limitations, initiatives to facilitate the clinical translation of nano-enabled technologies have been undertaken, such as the National Nanotechnology Initiative (NNI), proposed by the USA National Science and Technology Council (NSTC), which outlined well-defined initiatives and challenges for clinical translation [132].

## 5. Other Approaches in AMP Therapeutic Development

Although delivery systems based on nanomaterials have been widely applied to overcome AMP limitations, other technologies, such as chemical modifications, can be used [133]. Chemically modified AMPs can reduce barriers, including enzyme or salt degradation and rapid clearance, which cause low stability, bioavailability, and short half-life. Also, chemical modifications can overcome AMP cytotoxicity and poor membrane permeability due to size and hydrophobicity. These features impact the absorption, distribution, metabolism, and excretion properties of AMPs [20].

Among these chemical modifications, the substitution of one or more amino acid residues is widely used to optimize the physical-chemical AMP features [20]. Another chemical modification used is the addition of residues in the D-amino acid isoform (D-AAs) [14], since proteinases exclusively recognize L-amino acids (L-AAs), causing the degradation of peptides that contain residues with this configuration [133]. Cyclization of AMPs is also widely used to improve the antimicrobial activity and stability of the peptide, as well as to reduce its cytotoxicity [14,134].

N- and C-terminal modifications are also widely used to modify and improve the AMP properties. Acetylation at the N-terminus of the peptide increases its helicity and prevents enzymatic degradation. On the other hand, amidation at the C-terminus increases the stability and antimicrobial activity of the peptide [133]. Modifications such as adding fatty acids can increase the hydrophobicity of AMPs and their affinity for cell membranes. They can also hide regions of the AMP that would be vulnerable to protease attack. Therefore, the peptide andricin B was modified by adding a fatty acid at the N-terminus. Modified andricin B exhibited a 16-fold increase in antibacterial activity when compared to the peptide alone and also increased the stability of the peptide against proteases [135].

Another strategy to overcome the challenges related to the clinical application of AMPs is hybrid approaches [24]. The conjugation of AMPs with antibiotics may increase antimicrobial activity due to the synergistic effect and broader spectrum of activity. AMPs also facilitate the penetration of antibiotics into bacterial cells by interacting with bacterial cell membranes. Conjugation with AMPs can increase the specificity of antibiotics about bacterial cells, reducing toxicity effects. The increased antibacterial activity leads to the use of lower doses, reducing potential side effects and toxicity [24]. Some examples of hybrid approaches between AMPs and antibiotics in conjunction with nanoparticles are described here in this review. It is important to mention that various types and numbers of AMPs can be conjugated with each other and conjugated with antibiotics through appropriate ligands, allowing an effective treatment against several bacterial infections. Some examples of these hybrid approaches with nanoparticles are also described here in this review.

## 6. Conclusions and Future Perspectives

Bacterial resistance to conventional antimicrobials is a growing threat to global public health as it compromises the effectiveness of preventing and treating various infections. In that view, the search for alternative therapies has increased considerably. AMPs have been considered a promising therapeutic approach to combat bacterial infections. These molecules exert antibacterial activity through different mechanisms of action. However, AMPs have some intrinsic properties that limit their clinical application, such as their instability and toxicity. One of the approaches that can be applied to overcome these challenges related to AMPs is nanomaterial-based delivery systems. Nanoformulation holds significant potential to protect AMPs from adverse conditions such as degradation by proteases, pH changes, clearance, and neutralization by nonspecific binding promoting AMP stability and efficacy, enabling them to reach the site of infection, maximizing their effectiveness while minimizing systemic side effects. Furthermore, the nanoparticles themselves can exert antibacterial activity. This characteristic is very relevant because, in addition to protecting the AMPs, they can act in synergy with them, enhancing their effectiveness against bacterial infection. The clinical application of AMPs through nanoformulation has progressed significantly in recent years, and various AMPs were efficiently encapsulated and distributed by inorganic, polymer-based, and lipid-based NPs and gels, as described in this review. However, even with the nanoformulation of AMPs, there are still barriers that prevent their clinical application. Often, the results obtained in vitro assays are not reproduced in vivo assays, or there is the low efficiency of AMP encapsulation or its release from nanoparticles. To overcome these challenges, study design is essential. The physicochemical properties strongly influence the type of material with which they can be functionalized. Therefore, a rigorous study is necessary to choose the best delivery system for the target AMP. The interactions between the nanomaterials and the AMPs must be perfect so that the AMP is soluble, stable, and reaches the target site of infection without causing undesirable side effects. It is also necessary to carefully study the target site of infection and the possible interactions of the nanoencapsulated AMPs with the environment and thus select the best route of administration. System performance parameters should also be analyzed, such as formulation parameters and composition of the nanoparticles, release, and interaction with the target. These points can allow progress in this area, developing efficient and safe nanoparticle systems for AMP delivery, allowing these molecules to enter the clinical phases of drug development and be clinically applied to treat infectious diseases.

## Figures and Tables

**Figure 1 antibiotics-13-01042-f001:**
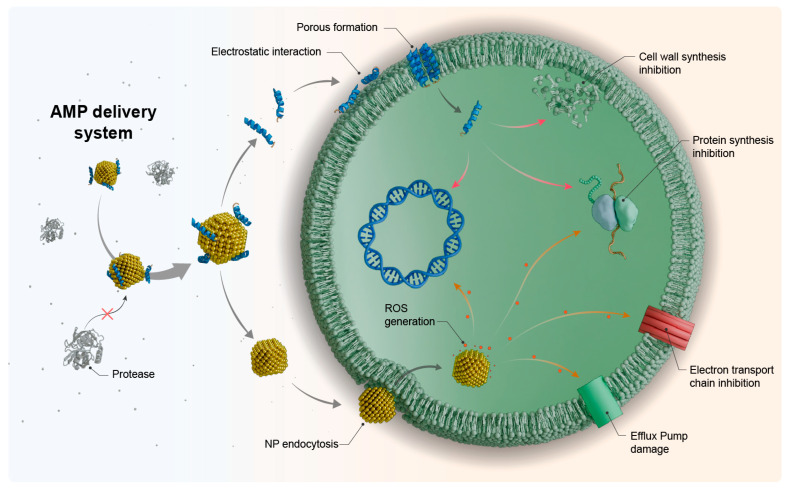
Mechanism of action of nanoformulated AMPs. Nanomaterials function as a delivery system for AMPs, providing a synergistic effect due to the impact of AMPs in conjunction with the effect of the nanomaterial itself, increasing antibacterial efficacy. AMPs can exert their antibacterial effect by destroying cell membranes through pores or carpet formation. AMPs can also cross the cell membranes and inhibit the synthesis or activity of intracellular molecules. Nanomaterials can protect the AMP from environmental conditions such as protease degradation and be antibacterial by destroying cell membranes and components such as efflux pumps and electron transport chains. Also, nanomaterials can cross the cell membranes and damage intracellular components by oxidative stress from ROS production.

**Figure 2 antibiotics-13-01042-f002:**
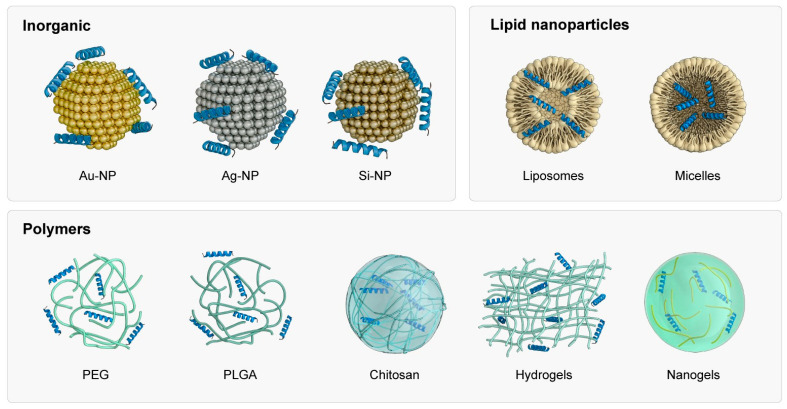
Different nanoformulations of AMPs as delivery systems. Inorganic nanoparticles include silver (Ag-), gold (Au-), and silicon (Si-) NPs. Lipid nanoparticles include liposomes and micelles. Polymeric nanoparticles can be made of polyethylene glycol (PEG), poly(lactic-co-glycolic) acid (PLGA), chitosan, hydrogels, and nanogels.

**Table 1 antibiotics-13-01042-t001:** AMPs conjugated to metal nanoparticles, including information about target infection and synergistic activity achieved.

AMP Nanoformulation	Target Bacteria	Synergistic Activity Achieved	Reference
Lys AB2 P3-His-Au-NPs	*A. baumanii*	Increase in the survival rate of infected mice and reduction in the cytotoxicity.	[38]
LL-37-Au-NPs	*S. aureus*	Increase wound healing.	[39]
Ura56-PEG-Au-NPs	MRSA, *E. coli*, multidrug-resistant *P. aeruginosa*, and *A. baumannii*	Increase in AMP stability. Increase in AMP efficacy. Reduction in the cytotoxicity.	[35]
Tryasine-Ag-NPs	*E. coli*, *S. aureus*	Increase in AMP efficacy. Reduction in the cytotoxicity.	[43]
(LLRR)3-Ag-NPs	*E. coli* and *S. aureus*	Increase in AMP efficacy. Reduction in the cytotoxicity.	[44]
Mellitin-ofloxacin-MSNs	*P. aeruginosa* biofilm	Increase in antibiofilm efficacy in vitro and in vivo.	[54]
NapFab-MSNs	*M. tuberculosis*	Increase in antibacterial efficacy.	[49]

**Table 2 antibiotics-13-01042-t002:** AMPs conjugated to polymeric nanoparticles, including information about target infection and synergistic activity achieved.

AMP Nanoformulation	Target Bacteria	Synergistic Activity Achieved	Reference
N6-PEG	*E. coli* and *Salmonella pullorum*	Increase in AMP stability. Increase in antibacterial efficacy. Increase in AMP biodistribution. Increase in AMP half-life.	[61]
KR12-PEG	*E. coli* biofilm	Increase in antibiofilm efficacy.	[62]
OH-CATH30-PLGA	*E. coli*	Increase in antibacterial efficacy. Increase in immunomodulatory activity.	[68]
NZ2114-PLGA	*Staphylococcus epidermidis*	Increase in antibacterial efficacy. Increase in the AMP stability. Reduction in cytotoxicity.	[69]
Octominin-chitosan-NPs	*Candida albicans* and *A. baumannii*	Increase in antibacterial efficacy. Reduction in cytotoxicity.	[75]
Cecropin-B-chitosan-NPs	MDR *K. pneumoniae*	Increase in antibacterial efficacy. Reduction in toxicity.	[76]
Jelleine-1 hydrogel	*E. coli* and *S. aureus*	Increase in antibacterial efficacy.	[81]
GelMA@Ti3C2/V-Os hydrogel	*S. aureus* and *E. coli*	Increase in antibacterial efficacy. Increase in immunomodulatory activity.	[78]
Novel AMP nanogel	*E. coli*, MRSA, and *P. aeruginosa*	Increase in antibacterial efficacy.	[86]
SAAP-148 and Ab-Cath-OL-HA nanogel	AMR *S. aureus* and *A. baumannii*	Improved selectivity. Reduced toxicity.	[89]

**Table 3 antibiotics-13-01042-t003:** Main AMP delivery systems, including their composition, strengths, weaknesses, and administration.

AMP Delivery System	Composition	Strengths	Weaknesses	Administration
Au-NPs	Gold	Biocapacity, relative stability, cell permeability. Antimicrobial, antioxidant, and anti-inflammatory activity. Reduces toxicity. Oxidative stress. Ease of industrial manufacturing for commercialization.	Oxidation-inducing toxicity. Accumulation in tissues. Poor biocompatibility. Lack of delivery ability.	Mainly topical delivery
Ag-NPs	Silver	Broad antibacterial action. Low toxicity. Oxidative stress. Strong bactericidal efficacy. Ease of industrial manufacturing for commercialization.	Accumulation in tissues. Poor biocompatibility. Lack of delivery ability.	Mainly topical delivery
MSNs	Silicon	Improves stability and bioavailability. Biocompatible. Biodegradable. Easier preparation and modification. High load capacity. Promotes controlled AMP loading and release. Ease of industrial manufacturing for commercialization.	Accumulation in tissues. Lack of delivery ability.	Mainly topical delivery
PEG	Small polymer particles	Improves solubility and bioavailability. Promotes controlled and sustained AMP release. Good colloidal integrity and stability. Bactericidal efficacy.	It is not biodegradable. Toxic organic preparation may generate residual material. It can cause immunogenicity. Can compromise AMP antimicrobial activity.	Mainly topical delivery
PLGA	Small polymer particles	Improves solubility and bioavailability. Promotes controlled and sustained AMP release. It is biodegradable. Reduces toxicity. Good colloidal integrity and stability. Bactericidal efficacy.	Toxic organic preparation may generate residual material. Can cause immunogenicity.	Mainly topical delivery
Chitosan	Small polymer particles	Improves solubility and bioavailability. Promotes controlled and sustained AMP release. Biocompatible. Biodegradable. Antimicrobial activity. Strong adhesion to the mucosa. Reduces toxicity. Good colloidal integrity and stability. Bactericidal efficacy.	Toxic organic preparation. may generate residual material. Can cause immunogenicity.	Mainly topical delivery
Hydrogels	Networks of crosslinked polymers with a high water content	Antimicrobial efficacy. Biocompatibility. No toxic organic preparation. Controlled and responsive AMP release. Good colloidal integrity and stability. Bactericidal efficacy.	Cell adhesion absence. Mechanical strength for some specific hydrogels.	Topical delivery, mainly wound healing
Liposomes	Vesicles of amphiphilic phospholipids and cholesterol	Improves biocompatibility, solubility, bioavailability, and reduces toxicity. Biodegradable. High load capacity. High encapsulation efficiency. Enhanced release. Ease of industrial manufacturing for commercialization.	Risk of phagocytosis and clearance. Polymeric changes and premature AMP release. Poor stability for long-term storage. Relatively weaker antibacterial activity.	Topical, oral, pulmonary, and systemic delivery
Micelles	Spheres of single-layer lipid vesicles of surfactants	Improves biocompatibility, solubility, bioavailability, and reduces toxicity. Biodegradable. High load capacity. Ease of industrial manufacturing for commercialization.	Less incorporation of hydrophobic AMPs. Polymeric changes and premature AMP release. Poor stability for long-term storage. Relatively weaker antibacterial activity.	

## Data Availability

No new data were created or analyzed in this study.

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
