# Peer review of "Antimicrobial Peptide Delivery Systems as Promising Tools Against Resistant Bacterial Infections"

_antibiotics, 2024, doi:10.3390/antibiotics13111042_

Round 1
Reviewer 1 Report
Comments and Suggestions for Authors
The manuscript (antibiotics-3220332) presents a comprehensive overview of antimicrobial peptide delivery systems as promising tools for combating resistant bacterial infections. Although, it covered significant amount of literature but it still needs improvement in several areas as mentioned in the specific comments below:
1. Authors should include the critical analysis regarding limitations linked with translation of nanoparticle conjugates on to clinical side.
2. Authors should incorporate the examples regarding the synergistic effects of employing nanoparticles with AMPs.
3. Do other approaches, such as chemical modification or hybrid approaches may be useful in overcoming limitations of clinical use of AMPs. If yes, please discuss in the relevant section.
4. The discussion regarding effect of properties of nanoparticles, like size, charge, surface features, etc. on delivery of AMP should be included.
5. Authors should consider adding more figures as involving relevant figures provides better perspective to the readers. Also, consider adding summary tables, for eg. Comparison of different AMP delivery systems, mechanisms, their applications, etc.
6. The possible limitation of utilizing the Au nanoparticles with regards to large scale production and associated cytotoxicity should be included.
7. Authors should discuss the safety concerns and resistance development with regards to using silver nanoparticles.
8. The benefits of employing hybrid nanoparticles over the single-component nanoparticles should be provided.
9. The challenges regarding scalability, stability and cost of manufacturing of polymeric nanoparticles should be discussed.
10. The comparative analysis among various types of polymeric nanoparticles should be included.
11. Does selection of polymers have potential impact over the cellular uptake and release properties of loaded AMP? If yes, please explain in relevant section.
12. Authors should consider including more recent literature for references as older references may not truly reflect recent advancements in the field.
Comments on the Quality of English Language
The manuscript needs to be edited to avoid inclusion of redundant sentences which reflects the similar desciption.
Also, the sentence formation and language edition should be done to improve readability.
Author Response
Dear Dr. Maresca and Dr. Simões,
Please find enclosed the response to the reviewer’s letter for the review entitled “Antimicrobial peptide delivery systems as promising tools against resistant bacterial infections” by Kamila B. S. Oliveira, Michel L. Leite, Nadielle T. M. Melo, Letícia F. Lima, Talita C. Q. Barbosa, Nathalia L. Carmo, Douglas A. B. Melo, Hugo C. Paes, and Octávio L. Franco that we are resubmitting to the Conference Special Issue "Rise of Antibiotic Resistance: Mechanisms Involved and Solutions to Tackle It” in the Antibiotics Journal. We appreciate and agree with the reviewers' suggestions. Original comments are in italics, and our answers are in regular typeface. All modifications are highlighted in the modified version of the manuscript. We believe the manuscript is improved after the changes suggested by the editorial board.
RESPONSES TO REVIEWERS’ COMMENTS
Reviewer 1. The manuscript (antibiotics-3220332) presents a comprehensive overview of antimicrobial peptide delivery systems as promising tools for combating resistant bacterial infections. Although, it covered significant amount of literature but it still needs improvement in several areas as mentioned in the specific comments below:
Major Comments:
1 Authors should include the critical analysis regarding limitations linked with translation of nanoparticle conjugates on to clinical side.
Response: The authors strongly agree with the reviewer's suggestions. Therefore, to address the limitations regarding the clinical translation of nanoparticle-based delivery systems, we have added a new topic, briefly discussing the main limitations regarding translation, as follows on pages 31-32:
“4. Clinical translation of nano formulated peptides
Liposomal doxorubicin (Doxil®), used for cancer treatment, was the first nanoparticle-based drug launched on the USA market in 1995 [1]. Since then, more than 30 nanoformulations have already been approved, and more than 100 clinical trials are underway for the application of nanoparticle-based therapies, demonstrating the enormous therapeutic potential of NP-based therapies [2]. Among the nanoparticle-based drugs approved by various agencies in both Europe and the USA, there are several to treat different types of cancer, as well as iron deficiency anemia, ultrasound contrast agents, and, more recently, the development of two vaccines for COVID-19 based on lipid nanoparticles [3].
With their promising potential, peptides offer a beacon of hope in overcoming inherent limitations that can impede the clinical translation of antimicrobial agents, particularly in the context of antisepsis. Their ability to enhance stability and cellular internalization (Gafar et al., 2024) is a significant stride forward. The unique physicochemical characteristics of peptides enable precise targeting, thereby reducing off-target effects. When combined with nanoparticles, these conjugates can be designed to be biocompatible and stable, thereby improving their pharmacokinetics and paving the way for clinical translation [4].
However, despite the increasing number of annual publications on nanoparticle drug-delivery systems, there are still many challenges to translating nanomedicine into clinical practice [5]. Limitations in the translation of NPs for clinical use occur due to the need for nanopharmaceutical design, which must necessarily consider the physical and chemical NP stability, its biodegradability, sophisticated formulation, and route of administration; in other words, it is necessary to know the in vitro and in vivo effects, biodistribution, pharmacokinetics, and pharmacodynamics of NPs . [6,7]. Large-scale production of NPs is another obstacle in the clinical translation of nanomedicine, as reproducibility is needed without a high production cost. In addition, polydispersity, scale-up complexity, incomplete contaminants purification, consistency and storage stability of the final product, morphology, and charge are other obstacles that need to be overcome [7].Clique ou toque aqui para inserir o texto.
In addition, there are other short-term challenges, such as: (i) our current understanding of the pathogenesis of infectious diseases is not sufficiently deep, making it difficult to act in rapidly changing conditions; (ii) the vast gap between animal models and humans; and (iii) the need for improvements in current technology for clinical design and translation [8]. They are associated with all these issues, properties of PNCs such as their size, shape, net surface charge, peptide characteristics, and conjugation chemistry can significantly limit their specificity and accuracy in theranostics [4].
Safety and regulatory compliance implications and the unique properties and potential risks of NPs limit their application in clinical use, as extensive testing and evaluation are mandatory before nanoparticle-based drugs are approved for clinical use [9]. Although there are many limitations, initiatives to facilitate the clinical translation of nano-enabled technologies have been undertaken, such as the National Nanotechnology Initiative (NNI), proposed by the USA National Science and Technology Council (NSTC), which outlined well-defined initiatives and challenges for clinical translation [10].”
2 Authors should incorporate the examples regarding the synergistic effects of employing nanoparticles with AMPs.
Response: The authors agree with the reviewer's suggestions. We have added a topic discussing the synergism of nano formulated peptides, as follows on pages 30-31:
“3. Synergistic effect of nanoformulated peptides.
Silver and zinc oxide nanoparticles release metal ions, which can act through electrostatic interactions with bacterial membranes, destabilizing the intracellular redox balance and causing DNA damage [11]. Similarly, AMPs have activity against a broad spectrum of microorganisms and are essential for the innate and acquired immune system, functioning as a defense mechanism. Because most active AMPs are cationic, they also act on the bacterial plasma membrane, including that of multidrug-resistant (MDR) bacterial strains, leading to membrane depolarization and cell permeabilization [12]. However, both NPs and AMPs have limitations that need to be overcome. Silver nanoparticles, for example, in addition to having low stability in aqueous systems, are prone to aggregation and can be cytotoxic to healthy cells [13]. Otherwise, AMPs may have underlying systemic toxicity [11]. Therefore, new strategies to overcome these limitations need to be developed.
In this scenario, the development of nanoparticle-based systems, which enable the controlled release of drugs or natural substances (such as peptides), is promising strategies to improve the antimicrobial activity of these compounds [14]. Biomolecules such as proteins, polysaccharides, and peptides with –OH, –COOH, and –NH2 functional groups can bind to metal nanoparticles through intermolecular hydrogen bonds, adsorbing onto their surfaces. This interaction increases silver nanoparticle stability in solution [13]. Another example is mesoporous silica nanoparticles (MSNs), which have advantages as drug carriers such as high porosity, low toxicity, and relative ease of chemical modification on their surface. These characteristics allow the development of complex encapsulated systems that can deliver the cargo on command by applying a specific external stimulus [14].
Recently, there has been an increase in publications demonstrating the improvement of the nanoformulation peptides' antimicrobial activity due to synergism [15–17]. It was observed that chitosan/bioactive glass nanoparticles/tetracycline composite coatings coated on the etched substrate and, with the subsequent addition of melittin by dripping, eradicated adherent bacteria and prevented biofilm formation on the implant surface. This fact demonstrates that there is a synergism between the melittin peptide and tetracycline when in a chitosan/bioactive glass coating nanoformulation, which can be explored as a multifunctional coating for bone implants [18]. Furthermore, the synergistic effect can be exploited for treating burn wound infections as demonstrated by Wali et al. in a study in which a combined dressing of decellularized human amniotic membrane (dHAM) loaded with colistin and AgNP was developed. The results indicated that dHAM associated with colistin and AgNP had a better performance in combating P. aeruginosa and K. pneumoniae when compared to the control in infected rats [19].
The combination of peptide, nanoparticle, and a conventionally used antibiotic can also be explored as an anti-biofilm. The IDR1018 peptide, when nanoformulated with chitosan nanoparticles (CNs) and in combination with ciprofloxacin showed an increased antibacterial and antibiofilm potential against clinical isolates of uropathogenic Escherichia coli (UPEC) resistant to ciprofloxacin [20]. Another approach is to exploit the synergism of peptides and nanoparticles to perform dual functions, as demonstrated by Wu et al. They developed zinc oxide (ZnO) nanoparticles with the glucagon-like peptide-1 (GLP-1) analog liraglutide (LG). ZnO is known to exert antibacterial activity against several strains, while LG acts to promote vascularization and wound healing. ZnO nanoparticles (ZnO-NPs) and LG were shown to simultaneously induce antibacterial, hemostatic, and vascularization effects for the healing of infected wounds [21].
Furthermore, the association between peptides and nanoparticles can decrease AMPs' cytotoxicity against healthy mammalian cells without affecting their antibacterial activity. Chitosan derivatives are capable of attaching different dendrimers of the G3K peptide and were efficient in eradicating P. aeruginosa cells for more than 24 h, in addition to reducing the cytotoxic effect, demonstrating synergism [22]. Another approach that can be taken is using the peptide as a matrix material for the tips of the microneedles, functioning as an excipient for drug delivery [17]. Recently, the peptide ε-poly-L-lysine (EPL) was used to produce dissolving microneedles that facilitate the intracellular accumulation of the antibiotic doxycycline (DOX) by increasing the permeability of the bacterial cell membrane. Using models of deep culturing infection induced by Pseudomonas aeruginosa, the EPL microneedles exhibited synergism, increasing the antimicrobial activity and prolonging the retention of DOX in the infected lesions [16]. A 99.91% reduction in bacterial load was observed in a single administration [16]. All the examples listed above show nanoformulated peptides' potential to treat different diseases. However, some limitations make translating nanoformulated peptides into the clinic difficult.”
3 Do other approaches, such as chemical modification or hybrid approaches may be useful in overcoming limitations of clinical use of AMPs. If yes, please discuss in the relevant section.
Response: We agree with the reviewer's comment. In that view, we have included a new section in the manuscript entitled “Other approaches in AMP therapeutic development” to discuss different approaches besides nanoencapsulation that can be applied to overcome AMP limitations, as follows on pages 32-33:
“5. Other approaches in AMP therapeutic development
Although delivery systems based on nanomaterials have been widely applied to overcome AMP limitations, other technologies, such as chemical modifications, can be used. [23]. Chemically modified AMPs can reduce barriers, including enzyme or salt degradation and rapid clearance, which cause low stability, bioavailability, and short half-life. Also, chemical modifications can overcome AMP cytotoxicity and poor membrane permeability due to the size and hydrophobicity. These features impact the absorption, distribution, metabolism, and excretion properties of AMPs. [24].
Among these chemical modifications, the substitution of one or more amino acid residues is widely used to optimize the physical-chemical AMP features [24]. Another chemical modification used is the addition of residues in the D-amino acid isoform (D-AAs) [25], since proteinases exclusively recognize L-amino acids (L-AAs), causing the degradation of peptides that contain residues with this configuration [26]. Cyclization of AMPs is also widely used to improve the antimicrobial activity and stability of the peptide, as well as to reduce its cytotoxicity [25,26].
N- and C-terminal modifications are also widely used to modify and improve the AMP properties. Acetylation at the N terminus of the peptide increases its helicity and prevents enzymatic degradation. On the other hand, amidation at the C terminus increases the stability and antimicrobial activity of the peptide [23]. Modifications such as adding fatty acids can increase the hydrophobicity of AMPs and their affinity for cell membranes. They can also hide regions of the AMP that would be vulnerable to protease attack. Therefore, the peptide andricin B was modified by adding a fatty acid at the N-terminus. Modified andricin B exhibited a 16-fold increase in antibacterial activity when compared to the peptide alone and also increased the stability of the peptide against proteases [27].
Another strategy to overcome the challenges related to the clinical application of AMPs is hybrid approaches [28]. The conjugation of AMPs with antibiotics may increase antimicrobial activity due to the synergistic effect and broader spectrum of activity. AMPs also facilitate the penetration of antibiotics into bacterial cells by interacting with bacterial cell membranes. Conjugation with AMPs can increase the specificity of antibiotics about bacterial cells, reducing toxicity effects. The increased antibacterial activity leads to the use of lower doses, reducing potential side effects and toxicity [28]. Some examples of hybrid approaches between AMPs and antibiotics in conjunction with nanoparticles are described here in this review. It is important to mention that various types and numbers of AMPs can be conjugated with each other and conjugated with antibiotics through appropriate ligands, allowing an effective treatment against several bacterial infections. Some examples of these hybrid approaches with nanoparticles are also described here in this review.
4 The discussion regarding effect of properties of nanoparticles, like size, charge, surface features, etc. on delivery of AMP should be included.
Response: The authors agree with the reviewer's suggestions. Paragraphs discussing the effect of nanoparticle properties such as size, charge, and surface characteristics have been already added to new the topic “4. Clinical translational of nanoformulated peptides”, as asked in suggestion 1, as follows on page 31-32.
5 Authors should consider adding more figures as involving relevant figures provides a better perspective to the readers. Also, consider adding summary tables, for eg. Comparison of different AMP delivery systems, mechanisms, their applications, etc.
Response: We strongly agree with the reviewer's comment. In that view, we have included one new figure summarizing the action mechanism of AMP conjugated with nano delivery systems to provide a better perspective to the readers on the manuscript, as follows on page 4:
Figure 1. Mechanism of action of nanoformulated AMPs. Nanomaterials function as a delivery system for AMP, providing a synergistic effect due to the impact of AMP in conjunction with the effect of the nanomaterial itself, increasing antibacterial efficacy. AMPs can exert their antibacterial effect by destroying cell membranes through pores or carpet formation. AMPs can also cross the cell membranes and inhibit the synthesis or activity of intracellular molecules. Nanomaterials can be antibacterial by destroying cell membranes and components such as efflux pumps and electron transport chains. Also, nanomaterials can cross the cell membranes and damage intracellular components by oxidative stress from ROS production.
We also included a summary table comparing the mainly AMP delivery systems, highlighting their composition, strengths, weaknesses, and administration already described in the manuscript, as follows on pages 25-29:
Table 3. Mainly AMP delivery systems, including their composition, strengths, weaknesses, and administration.
AMP delivery system |
Composition |
Strengths |
Weaknesses |
|
Administration |
|
AuNPs |
Gold |
Biocapacity, relative stability, cell permeability Antimicrobial, antioxidant, and anti-inflammatory activity Reduce toxicity Oxidative stress Ease of industrial manufacturing for commercialization |
Oxidation-inducing toxicity Accumulation in tissues Poor biocompatibilitie Lack of delivery ability |
|
Mainly topical delivery |
|
AgNPs |
Silver |
Broad antibacterial action Low toxicity Oxidative stress Strong bactericidal efficacy Ease of industrial manufacturing for commercialization
|
Accumulation in tissues Poor biocompatibilitie Lack of delivery ability |
|
Mainly topical delivery |
|
MSNs |
Silicon |
Improve stability and bioavailability. Biocompatible Biodegradable Easier preparation and modification High load capacity Promote controlled AMP loading and release Ease of industrial manufacturing for commercialization
|
Accumulation in tissues Lack of delivery ability |
|
Mainly topical delivery |
|
PEG |
Small polymer particles |
Improve solubility and bioavailability Promote controlled and sustained AMP release Good colloidal integrity and stability Bactericidal efficacy |
It is not biodegradable Toxic organic preparation may generate residual material It can cause immunogenicity. Can compromise AMP antimicrobial activity |
|
Mainly intravenous |
|
PLGA |
Small polymer particles |
Improve solubility and bioavailability. Promote controlled and sustained AMP release It is biodegradable Reduce toxicity Good colloidal integrity and stability Bactericidal efficacy
|
Toxic organic preparation may generate residual material Can cause immunogenicity |
|
Mainly intravenous |
|
Chitosan |
Small polymer particles |
Improve solubility and bioavailability. Promote controlled and sustained AMP release Biocompatible Biodegradable Antimicrobial activity Strong adhesion to the mucosa Reduce toxicity Good colloidal integrity and stability Bactericidal efficacy |
Toxic organic preparation may generate residual material Can cause immunogenicity |
|
Mainly topical delivery |
|
Hydrogels |
Networks of crosslinked polymers with a high-water content |
Antimicrobial efficacy Biocompatibility No toxic organic preparation Controlled and responsive AMP released Good colloidal integrity and stability Bactericidal efficacy
|
Cell adhesion absence Mechanical strength for some specific hydrogels |
|
Topical delivery, mainly wound healing |
|
Liposomes |
Vesicles of amphiphilic phospholipids and cholesterol |
Improve biocompatibility, solubility, bioavailability and reduce toxicity. Biodegradable. High load capacity. High encapsulation efficiency. Enhanced release. Ease of industrial manufacturing for commercialization.
|
Risk of phagocytosis and clearance. Polymeric changes and premature AMP release. Poor stability for long-term storage. Relatively weaker antibacterial activity.
|
|
Topical, oral, pulmonary, and systemic delivery |
|
Micelles |
Spheres of single-layer lipid vesicles of surfactants |
Improve biocompatibility, solubility, bioavailability and reduce toxicity. Biodegradable. High load capacity. Ease of industrial manufacturing for commercialization.
|
Less incorporation of hydrophobic AMPs. Polymeric changes and premature AMP release. Poor stability for long-term storage. Relatively weaker antibacterial activity.
|
|
|
|
We also have designed specific tables in each section, including descriptions of the nano-formulated AMPs, targets, and synergic activity to provide a better perspective to the readers, as follows on pages 10-11:
Table 1. AMPs conjugated to metal nanoparticles, including information about target infection and synergistic activity achieved.
AMP nano formulated |
Target bacteria |
Synergistic activity achieved |
Reference |
Lys AB2 P3-His-Au-NPs |
A. baumanii |
Increase in the survival rate of infected mice and reduction in the cytotoxicity |
[29]. |
LL-37-AuNPs |
S. aureus |
Increase the wound healing |
[30]. |
Ura56-PEG-Au-NPs |
MRSA, E. coli, multidrug-resistant P. aeruginosa, and A. baumannii |
Increase the AMP stability Increase the AMP efficacy Reduction in the cytotoxicity |
[31]. |
Tryasine-Ag-NPs |
E. coli, S. aureus |
Increase the AMP efficacy Reduction in the cytotoxicity |
[32]. |
(LLRR)3-Ag-NPs |
E. coli and S. aureus |
Increase the AMP efficacy Reduction in the cytotoxicity |
[13]. |
Mellitin- ofloxacin-MSNs |
P. aeruginosa biofilm |
Increase antibiofilm efficacy in vitro and in vivo
|
[33]. |
NapFab-MSNs |
M. tuberculosis |
Increase antibacterial efficacy |
[34] |
As follows on pages 19-20:
Table 2. AMPs conjugated to polymeric nanoparticles, including information about target infection and synergistic activity achieved.
AMP nano formulated |
Target bacteria |
Synergistic activity achieved |
Reference |
N6-PEG |
E. coli and Salmonella pullorum |
Increase the AMP stability Increase antibacterial efficacy Increase AMP biodistribution Increase AMP half-life |
[35] |
KR12-PEG |
E. coli biofilm |
Increase antibiofilm efficacy |
[36] |
OH-CATH30- PLGA |
E. coli |
Increase antibacterial efficacy Increase immunomodulatory activity |
[37] |
NZ2114-PLGA |
Staphylococcus epidermidis |
Increase antibacterial efficacy Increase the AMP stability Reduction in cytotoxicity |
[38] |
Octominin-chitosan-NPs |
Candida albicans and A. baumannii |
Increase antibacterial efficacy Reduction in cytotoxicity |
[39] |
Cecropin-B-chitosan-NPs |
MDR K. pneumoniae |
Increase antibacterial efficacy Reduce toxicity |
[40] |
Jelleine-1-hydrogel |
E. coli and S. aureus |
Increase antibacterial efficacy |
[41] |
GelMA@Ti3C2/V-Os hydrogel |
S. aureus and E. coli |
Increase antibacterial efficacy Increase immunomodulatory activity |
[42] |
Novel AMP nanogel |
E. coli, MRSA, and P. aeruginosa |
Increase antibacterial efficacy |
[43] |
SAAP-148 and Ab-Cath-OL-HA nanogel |
AMR S. aureus and A. baumannii |
Improve selectivity Reduce toxicity |
[44] |
6 The possible limitation of utilizing the Au nanoparticles with regards to large-scale production and associated cytotoxicity should be included.
Response: We agree with the reviewer's suggestion. We have already described the nanoparticle's limitations in general including large-scale production in the new topic “4. Clinical translational of nanoformulated peptides”, as asked in suggestion 1, as follows on pages 31-32. We have included a paragraph briefly describing the cytotoxicity limitation regarding metal nanoparticles in general as follows on page 8, lines 248-258:
“Another issue of metal NPs is their unpredictable toxic effects that threaten human health. The cytotoxicity of metal NPs is directly associated with their properties such as size, shape, composition, charge, solubility, and coating material. Metal NPs can penetrate cells and interact with other molecules such as proteins, causing neurotoxicity, immunotoxicity, and genotoxicity. Oxidative stress is one of the main mechanisms of cytotoxicity of metal NPs, caused by the excessive production of reactive oxygen species (ROS), which alters the oxidation-reduction state. Another mechanism is inflammation, a natural protective response to infection that can have detrimental effects if not regulated. Both mechanisms are related. Therefore, the biosafety of these materials is a major issue that needs to be solved for their wide clinical application [45].”
7 Authors should discuss the safety concerns and resistance development with regards to using silver nanoparticles.
Response: We agree with the reviewer's suggestion. Therefore, we have included a paragraph describing the resistance development regarding silver nanoparticles, as follows on page 8, lines 259-270:
“The development of bacterial resistance is a natural process of adaptation and survival of bacteria. Bacteria have already demonstrated resistance against AgNPs through the secretion of the protein flagellin, which is capable of inducing the coagulation process of AgNPs and drastically reducing their antibacterial activity [46]. Strategies have been employed to overcome this challenge, such as binding silver nanoparticles to cyanographene. This combination was able to eliminate AgNP-resistant bacteria at a MIC value of 3.4 mg. L-1 against Ag-resistant E. coli compared to 108 mg. L-1 for AgNPs alone, 1.9 mg. L-1 against Ag-resistant P. aeruginosa compared to 54 mg. L-1 for AgNPs alone. The strong interaction between cyanographene and silver profoundly suppressed silver leaching [46]. Bacterial resistance against Au-NPs has also already been described, and strategies to overcome this issue have been proposed [47].”
8 The benefits of employing hybrid nanoparticles over the single-component nanoparticles should be provided.
Response: We agree with the reviewer's suggestion. Therefore, we have included a paragraph describing the advantages of hybrid nanoparticles, as follows on page 9, lines 290-299:
“Despite having many advantages for delivering AMPs, nanoparticles also present problems such as being phagocytosed or accumulating in the liver, which reduces their therapeutic efficacy and increases their toxic effect. Among the strategies that can be used to solve these problems, hybrid nanoparticles stand out. The use of different nanomaterials together, such as using another nanomaterial with inert or hydrophilic molecules to modify the surface of a nanoparticle, can overcome these problems [48,49], such as described above about binding silver nanoparticles to cyanographene. There are several possibilities for hybrid nanoparticles, such as polymer-metal, graphene oxide-metal, graphene oxide–polymer and lipid polymer hybrid nanoparticles [50]. Some examples of hybrid nanoparticles are described in the manuscript”
9 The challenges regarding scalability, stability and cost of manufacturing of polymeric nanoparticles should be discuss.
Response: We agree with the reviewer's suggestion. Therefore, we have already discussed the issues related to scalability, stability, and cost of manufacturing nanoparticles in general in the new topic “4.Clinical translation of nano-formulated peptides” included in the manuscript, as follows on pages 31-32. We have included a brief paragraph regarding the benefits of polymeric nanoparticles regarding costs, as follows on page 11, lines 361-365:
“The easy synthesis of polymers in large quantities and at much lower costs are very attractive features for nanoformulating AMPs. The polymers also have a broad spectrum of antibacterial action and a low propensity for the development of bacterial resistance. Furthermore, they present greater stability in biological and storage conditions and controlled release [50]”
10 The comparative analysis among various types of polymeric nanoparticles should be included.
Response: We agree with the reviewer's suggestion. Therefore, we have included more information regarding polymeric nanoparticles highlighting the advantages and disadvantages of each one, as follows on page 11, lines 358-365:
“Polymers also enable controlled and sustained release of AMPs, and site-specific accumulation, thereby reducing dosage [51,52]. AMPs can be adsorbed, dissolved, encapsulated, or even attached to the polymer, being released mainly through diffusion [53].”
as follows on page 11, lines 370-371:
“and water solubility. PEG conjugation also promotes lower renal clearance [54].”
as follows on page 12, lines 407-412:
“It is important to emphasize that despite PEG improving the stability of protease, and prolonged action time of AMPs, this material is not biodegradable and can accumulate in human organs and tissues, causing immunogenicity and limiting its therapeutic use. Some reports have shown that PEG can reduce peptide efficacy. Therefore, the treatment of bacterial infections with AMPs associated with PEG is still a challenge in vivo [54,55].”
as follows on page 14, lines 485-492:
“activity and low toxicity. Chitosan has a cationic structure due to the presence of positive amino groups, which enables the interaction with negative compounds such as bacteria membranes [53]. It also protects AMPs from degradation during administration and sustains their release, thus increasing bioavailability [51,56]. Also, it has a low cost. Chitosan is specially used for wound healing, promoting wound closure rate, neovascularization, re-epithelialization, and cellular proliferation at the location of the wound [53].”
as follows on page 15, lines 529-531:
“Hydrogels do not need toxic organic solvents in preparation, avoiding residual material and have controlled and responsive drug release [57].”
as follows on page 15, lines 533-535:
“The polymers used here can be natural such as hyaluronic acid, hyaluronic acid, and alginate; or synthetic as poly (ethyl acrylate-co-methacrylic acid) and poly (Nisopropylacrylamide)] [53].”
as follows on page 17, lines 605-606:
“Hydrogels are classified as macroscopic hydrogels, nanogel, and microgels, according to their size [53].”
as follows on page 21, lines 677-679:
“Despite the considerable strengths of polymeric nanoparticles, they present some drawbacks such as the toxic organic preparation and degradation, and the residual material, causing immunogenicity [57].”
11 Does selection of polymers have potential impact over the cellular uptake and release
properties of loaded AMP? If yes, please explain in relevant section.
Response: We agree with the reviewer's suggestion. Material selection of nanoparticles in general, not only polymers, has a relevant impact on cellular uptake and release properties of loaded AMP. Therefore, we have included a paragraph briefly describing the main cellular uptake mechanisms of nanoparticles and their importance, as follows on page 30, lines 841-855:
“The choice of nanomaterial is critical since nanoparticles interact directly with the plasma membrane of cells and this is reflected in the uptake of nanoparticles. The plasma membrane comprises a phospholipid bilayer containing several biomolecules and is mostly negatively charged. Given this, it has selective permeability to ions, biomolecules, and nanoparticles [58]. Therefore, it is essential to take into account the chemical, physical, and structural characteristics of the nanomaterial, as well as the entry mechanism, since all of this determines the entry and performance of the function of the nanoparticle in question. Nanoparticles cross the plasma membrane by endocytosis-based absorption pathways and direct cellular entry. Endocytosis consists of different pathways and mechanisms mediated by lipids and transport proteins, enabling nanoparticles to cross cell membranes and enter cells. After being endocytosed, nanoparticles are stored in endosomes, phagosomes, or macropinosomes and do not have direct and rapid access to the cytoplasm or cell organelles. Direct access through delivery channels allows nanoparticles to access the entire cytoplasm and interact with intracellular organelles and structures, facilitating the performance of their specific biological functions [58].”
12 Authors should consider including more recent literature for references as older references may not truly reflect recent advancements in the field.
Response: We agree with the reviewer's suggestion. Therefore, all new references included in the manuscript are from 2020 until now as can be seen through the manuscript.
Reviewer 2 Report
Comments and Suggestions for Authors
The abstract is clear and offers an appropriate summary of the study. Its structure is sound, but it could be improved by incorporating more specific quantitative data.
The introduction effectively highlights the severity of antimicrobial resistance, supported by relevant data. The review proficiently outlines the potential of AMPs (antimicrobial peptides) to address these issues and introduces the concept of delivery systems clearly.
Here, I would suggest adding a paragraph to briefly explain how AMPs function, as some readers may not be familiar with their mechanism of action.
I recommend simplifying the terminology or providing brief definitions for highly technical terms to enhance readability (one paragraph or two).
Overall, the manuscript is clear and understandable.
I would propose adding some information comparing different delivery systems in terms of efficiency, safety, and clinical potential, even if it's just a short paragraph (one paragraph or two)..
Regarding references, the older ones, particularly those from the 1990s, should be minimized unless they provide fundamental information that remains highly relevant.
In my opinion, these are minor revisions, and the article is more than adequate for publication.
Comments on the Quality of English LanguageIn my opinion, these are minor revisions, and the article is more than adequate for publication.
Author Response
Dear Dr. Maresca and Dr. Simões,
Please find enclosed the response to the reviewer’s letter for the review entitled “Antimicrobial peptide delivery systems as promising tools against resistant bacterial infections” by Kamila B. S. Oliveira, Michel L. Leite, Nadielle T. M. Melo, Letícia F. Lima, Talita C. Q. Barbosa, Nathalia L. Carmo, Douglas A. B. Melo, Hugo C. Paes, and Octávio L. Franco that we are resubmitting to the Conference Special Issue "Rise of Antibiotic Resistance: Mechanisms Involved and Solutions to Tackle It” in the Antibiotics Journal. We appreciate and agree with the reviewers' suggestions. Original comments are in italics, and our answers are in regular typeface. All modifications are highlighted in the modified version of the manuscript. We believe the manuscript is improved after the changes suggested by the editorial board.
RESPONSES TO REVIEWERS’ COMMENTS
Reviewer 2.
Minor Comments:
1 The abstract is clear and offers an appropriate summary of the study. Its structure is sound, but it could be improved by incorporating more specific quantitative data.
Response: The authors agree with the reviewer. In that view, we incorporated specific quantitative data into the abstract, as follows on page 1, lines 19-22:
“It is estimated that around 1.27 million people died worldwide in 2019 due to infectious diseases caused by antibiotic-resistant microorganisms, according to the WHO. It is estimated that 700,000 people die each year worldwide, which is expected to rise to 10 million by 2050.”
Information from https://www.who.int/news-room/fact-sheets/detail/antimicrobial-resistance, accessed on 21.09.24.
2 The introduction effectively highlights the severity of antimicrobial resistance, supported by relevant data. The review proficiently outlines the potential of AMPs (antimicrobial peptides) to address these issues and introduces the concept of delivery systems clearly. Here, I would suggest adding a paragraph to briefly explain how AMPs function, as some readers may not be familiar with their mechanism of action. I recommend simplifying the terminology or providing brief definitions for highly technical terms to enhance readability (one paragraph or two). Overall, the manuscript is clear and understandable.
Response: The authors agree with the reviewer. In that view, we added some more information about the mechanism action of AMPs, to improve the understanding of the reader, as follows on page 2, lines 54-62:
“AMPs can combat bacteria through various mechanisms, acting on the cell wall of microbial cells, affecting the synthesis of essential components such as peptidoglycans, teichoic acid, lipoproteins, and lipopolysaccharide, destroying its structure. They can also affect the process of bacterial division (X. Liet al., 2022; Xuan et al., 2023). AMPs interact with bacterial cells through electrostatic interactions and gradually accumulate on the cell membrane surface until a threshold concentration is reached. After that, AMPs act through different modes of action, forming transmembrane channels involving rod pores or toroidal pores on the target cell membrane [59,60].”
As follows on page 2, lines 66-70:
“These mechanisms of action increase membrane permeability, leading to leakage of cell contents and cell death [25]. Furthermore, AMPs can cross the cell wall through direct penetration or endocytosis and inhibit the synthesis or activity of intracellular molecules, such as nucleic acids and intracellular proteins [61].”
We also describe more about the drawbacks of AMPs to be overcome, specifically about microbial resistance, since this is an obstacle that nanoparticles can also overcome. We also describe some examples of AMPs that have antibacterial solid activity but still have limitations for their clinical application. We believe that this information considerably improves the quality of the manuscript, as follows on pages 2, lines 76-77:
“AMPs also present challenges related to their production, such as high cost and large-scale production [28].”
as follows on pages 2-3, lines 84-93:
“Another relevant problem related to AMPs is that some bacteria are already resistant to AMPs due to the process of natural selection. Resistant bacteria can cause modifications in components of the membrane structure, neutralize AMPs through secreted proteases, or expel transmembrane AMPs through efflux pumps. There are already marketed AMP-based drugs that present disadvantageous, mainly considerable adverse reactions, which still limit their complete clinical application [62]. The commercial AMP vancomycin [63] and murepavadin [64] Have strong antibacterial activity but also present ototoxicity, nephrotoxicity, allergy, diarrhea, oral intolerance, inflammation, and renal toxicity as limitations [62].”
3 I would propose adding some information comparing different delivery systems in terms of efficiency, safety, and clinical potential, even if it is just a short paragraph (one paragraph or two).
Response: The authors agree with the reviewer. Reviewer 1 asked the same for polymeric nanoparticles, so they are already described above, as follows on pages 11-21. In that view, we added more information about the positive features of the inorganic and lipidic delivery systems, such as efficiency, safety, and clinical potential to complement the advantages already described. We also added some information related to the drawbacks of the different delivery systems to improve the quality of the manuscript, as follows on page 5, lines 146-152:
“Metal nanoparticles do not have a specific mechanism of action and do not bind to specific receptors on bacterial cells. This feature increases the spectrum of antibacterial activity and hinders the development of resistance by bacteria [52,55]. Among the various advantages that inorganic nanoparticles offer, we should mention the oxidative stress caused by the ROS production that can act on the disruption of cell walls and damage of DNA and RNA molecules or even their synthesis inhibition [65].”
Page 5, lines 153-155:
“Therefore, gold nanoparticles (Au-NPs) and silver nanoparticles (Ag-NPs) have been conjugated with AMPs, improving the effects of both molecules, potentially reducing the toxicity of the metal nanoparticles and increasing the AMP efficacy [54].”
As follows on page 8, lines 245-248:
“Despite the advantages mentioned here, some drawbacks related to metal nanoparticles need to be overcome. The therapeutic effects in mouse models, such as the bacterial load reduction and inflammatory damage in organs, are not usually as satisfactory as in vitro assays [52,55].”
As follows on page 8, lines 280-282:
“One of the great advantages of this system is that the well-defined pore structure allows control of drug loading and release kinetics, which prevents the degradation of AMPs by proteases. [55]”
As follows on page 21, lines 681-685:
“These are small spherical vesicles with high surface area and are composed of ionizable lipids. Lipic nanoparticles usually have high biocompatibility, bioavailability, biodegradability, solubility, and reduced toxicity. These nanoparticles include liposomes, micelles, and liquid crystalline nanoparticles. [57,66].”
As follows on page 21, lines 693-698:
“The dual polarity of liposomes due to the hydrophilic lipid head and hydrophobic tail allows the incorporation of hydrophilic and hydrophobic molecules [52]. Liposomes show high encapsulation efficiency, enhanced release, and antimicrobial effect [65]. Liposomes are biocompatible and biodegradable nanocarriers applied for topical, oral, pulmonary, and systemic delivery [53].”
As follows on page 22, lines 719-722:
“The advantages of liposomes for the delivery of AMPs are undeniable. However, these vectors still risk phagocytosis and clearance by the reticuloendothelial system. Strategies such as surface functionalization to induce charge inversion or size control can be used to overcome phagocytosis [62].”
As follows on page 22, lines 725-726:
“PEGylation of the surface of liposomes prolongs circulation time and reduces their uptake by macrophages [53].”
As follows on page 22, lines 736-738:
“However, these strategies can negatively impact the activity of liposomes, causing a decrease in the efficiency of immune cell recruitment, impairing the reduction of inflammation or the elimination of bacteria [62].”
As follows on page 22, lines 741-742:
“However, the incorporation of hydrophobic AMPs is limited due to the structure of the micelles [53].”
As follows on page 23, lines 771-773:
“This system has advantageous characteristics such as more excellent solubility, bioavailability, and stability [53].”
As follows on page 23, lines 779-780:
“Lipidic nanoparticles face other drawbacks such as polymeric changes and premature AMP release [57].”
4 Regarding references, the older ones, particularly those from the 1990s, should be minimized unless they provide fundamental information that remains highly relevant.
Response: The authors agree with the reviewer. The 1990 references in the manuscript are references that remain highly relevant, such as follows on page 2, lines 64-66:
“…or by the so-called Shai-Huang-Matsuzaki (SHM) model, whereby the peptides act through different mechanisms (Gazit et al., 1996; Ludtke et al., 1996; Matsuzaki et al., 1994).”
In my opinion, these are minor revisions, and the article is more than adequate for publication.

Reviewer 3 Report
Comments and Suggestions for Authors
My comments/suggestions are attached herewith.

Mostly okay and understandable. May require minor editing.
Author Response
Dear Dr. Maresca and Dr. Simões,
Please find enclosed the response to the reviewer’s letter for the review entitled “Antimicrobial peptide delivery systems as promising tools against resistant bacterial infections” by Kamila B. S. Oliveira, Michel L. Leite, Nadielle T. M. Melo, Letícia F. Lima, Talita C. Q. Barbosa, Nathalia L. Carmo, Douglas A. B. Melo, Hugo C. Paes, and Octávio L. Franco that we are resubmitting to the Conference Special Issue "Rise of Antibiotic Resistance: Mechanisms Involved and Solutions to Tackle It” in the Antibiotics Journal. We appreciate and agree with the reviewers' suggestions. Original comments are in italics, and our answers are in regular typeface. All modifications are highlighted in the modified version of the manuscript. We believe the manuscript is improved after the changes suggested by the editorial board.
RESPONSES TO REVIEWERS’ COMMENTS
Reviewer 3. This review article [“Antimicrobial peptide delivery systems as promising tools against resistant bacterial infections”] by Oliveira et al. outlines significant research trends regarding the clinical application of antimicrobial peptides (AMPs) against resistant bacterial infections through several previously established nanoformulation methods. The review looks interesting, is quite extensive. However, I have a few suggestions to the authors before I can recommend it for the publication in the journal Antibiotics and I think incorporating those would definitely improve the article. My suggestions are listed below:
Major Comments:
1 I believe that this review article can significantly benefit from having a few summary figures. Having such figures would definitely help the readers easily understand complex information, such as complex processes, or phenomena that are discussed in the text and also make the article more attractive. In that direction, I have a few suggestions:
On Page 2: About the different mechanism of action of the AMPS, adding a summary figure to depict the mechanisms discussed would definitely help. Again, when the authors discuss about nanomaterials themselves having the ability to control infections in vitro and in vivo, via a variety of mechanisms – I would recommend include an illustrative figure explaining them.]
Response: We strongly agree with the reviewer that adding new figures will improve the quality of our manuscript. We have designed a figure to illustrate this mechanism of action, emphasizing the different mechanisms of action by which AMPs and nanosystems exert together, since the focus of our review is to describe the benefits of synergism between AMPs and delivery systems. Reviewer 1 had also requested the addition of new graphic elements, so the new figure can be seen just above in this letter, as follows on page 4.
2 The article can also benefit from having a few summary tables. For example, it would be really nice to have table of different AMPs discussed in each sections, their sequences, source, target, activity, with reference?
Response: We strongly agree with the reviewer that adding a few summary tables will improve the quality of our manuscript. We have designed specific tables in each section, including descriptions of the nano-formulated AMPs, targets, and synergic activity since our review focuses on the benefits of synergism between AMPs and the delivery systems. Furthermore, we have included a table describing the main AMP delivery systems, composition, strengths, weaknesses, and administration. Reviewer 1 had also requested tables as graphical elements, so the added tables can be reviewed just above on this letter, as follows on pages 10-11, 19-20, 25-29.
3 In the conclusion section, a little more detailed discussion is needed regarding the strengths and drawbacks of each of the encapsulation techniques or delivery methods. This can also be done by including a summary table.
Response: We agree with the reviewer. We have included a summary table comparing the main AMP delivery systems highlighting their composition, strengths, weaknesses, and administration. Reviewer 1 had also requested the addition of new graphical elements, so the new table can be reviewed just above on this letter, as follows on pages 25-29.
Furthermore, we have rewritten the whole conclusion section describing our perspective associated with the challenges of AMP nanoformulation along with potential solutions and future perspectives, as follows on pages 33-34:
“Conclusion and future perspectives
Bacterial resistance to conventional antimicrobials is a growing threat to global public health as it compromises the effectiveness of preventing and treating various infections. In that view, the search for alternative therapies has increased considerably. AMPs have been considered a promising therapeutic approach to combat bacterial infections. These molecules exert antibacterial activity through different mechanisms of action. However, AMPs have some intrinsic properties that limit their clinical application, such as their instability and toxicity. One of the approaches that can be applied to overcome these challenges related to AMPs is nanomaterial-based delivery systems. Nanoformulation holds significant potential to protect
AMP from adverse conditions such as degradation by proteases, pH changes, clearance, and neutralization by nonspecific binding promotes AMP stability and efficacy, enabling them to reach the site of infection and maximizing their effectiveness while minimizing systemic side effects. Furthermore, the nanoparticles themselves can exert antibacterial activity. This characteristic is very relevant because, in addition to protecting the AMPs, they can act in synergy, enhancing their effectiveness against bacterial infection. The clinical application of AMPs through nanoformulation has progressed significantly in recent years, and various AMPs were efficiently encapsulated and distributed by inorganic, polymer-based, lipid-based NPs and gels as described in this review. However, even with the nanoformulation of AMPs, barriers still prevent their clinical application. Often, the results obtained in in vitro assays are not reproduced in vitro assays, or there is low efficiency of AMP encapsulation or its release from nanoparticles. To overcome these challenges, study design is essential. The physicochemical properties strongly influence the type of material with which they can be functionalized. Therefore, a rigorous study is necessary to choose the best delivery system for the target AMP. The interactions between the nanomaterials and the AMPs must be perfect so that the AMP is soluble, stable, and reaches the target site of infection without causing undesirable side effects. It is also necessary to carefully study the target site of infection and the possible interactions of the nanoencapsulated AMPs with the environment and thus select the best route of administration. System performance parameters should also be analyzed, such as formulation parameters and composition of the nanoparticles, release, and interaction with the target. These points can allow progress in this area, developing efficient and safe nanoparticle systems for AMP delivery, allowing these molecules to enter the clinical phases of drug development and be clinically applied to treat infectious diseases.”
- In the ‘Lipid nanoparticles’ section: This article is silent about the new class of lipid nanoparticles that have been used for targeted deliveries. One such examples is: “Gut targeted nanoparticles deliver specifically targeted antimicrobial peptides against Clostridium perfringens infections” by Xu et al.; DOI: 10.1126/sciadv.adf8782. I would encourage the authors to search for more articles in that direction and include them in the modified version of this review article.
Response: We agree with the reviewer. In that view, we have included the suggested reference in the modified version of our review article. The new class described in the suggested reference is mesoporous, not lipidic nanoparticles. Therefore, we cited the reference in the correct section, as follows on page 10, lines 336-342:
“A novel type of delivery system based on mesoporous fusiform nanoparticles, termed a gut-targeted engineered particle vaccine (EPV), has been designed for the targeted delivery of two hybrid AMPs, F6P1 and F6P6, to the intestine to treat Clostridium perfringens infection. The EPV delivery system was designed to increase the specificity and bioavailability of the antibacterial, antibiofilm hybrid AMPs (HAMPs). HAMPs conjugated with EPV exhibited greater antimicrobial activity against C. perfringens colonic infections compared to HAMPs alone according to MIC and antibiofilm assays [67].”
- In the ‘Hydrogels’ section: Please include the references and discussion about the most recent advancements. Two such examples are: 1)https://doi.org/10.1002/adhm.202401289, 2) https://doi.org/10.1021/acsami.3c00191.
Response: We agree with the reviewer. In that view, we included the suggestions references to the manuscript to point out the most recent advancements in the area, as follows on page 16-17, lines 581-604:
“A recent advancement in hydrogels is the redox-degradable hydrogel, used for polymer-based hydrogels. Disulfide bonds are introduced into a polymer structure and can be cleaved in response to variations in the environmental redox state, such as reactive oxygen species (ROS) produced in the wound, inflammation, or sites of bacterial infection and biofilms. Thus, there is stimulation of the release of the therapeutic agent and degradation of the hydrogel. The AMP vancomycin, a glycopeptide antibiotic medication, was conjugated in a redox-degradable hydrogel to treat skin infections topically to overcome the challenges of high doses needed for intravenous administration. The redox-degradable hydrogel loaded with vancomycin showed
an effective long-lasting antibacterial activity against E. coli. In vivo, the wound healing model assay showed that the percentage of wound contraction after 3 days of surgery was 58.65±15.1 for 8% hydrogel+vancomycin while for group control was 36.3 ± 18%, indicating the importance of redox-degradable hydrogel [68].
An unnamed peptide, it was also nanoformulated with a type of precisely controlled release hydrogel in response to environmental factors such as reactive oxygen species (ROS) and matrix metalloproteinases (MMPs) infection. The AMP was conjugated with a hydrogel composed of hyaluronic acid modified with cyclodextrin (HA−CD) and adamantane (Ad−HA). Ad-HA-AMP improved AMP stability and antimicrobial activity against E. coli and S. aureus with a live bacteria rate of less than 20%. The controlled release of AMP induced by the MMP and ROS promotes cell viability of more than 98%. In comparison, the control group caused cytotoxicity, decreasing cell viability by less than 80% due to uncontrolled release. Furthermore, the diabetic chronic wound model in vivo assays demonstrated that wound healing was improved as assessed by wound diameter (less than 35%) and the presence of bacteria (less than 10%) [69].”
- I would suggest that the authors include a discussion about the most recent advancements in combating multidrug-resistant bacteria with structurally nanoengineered antimicrobial peptide polymers (SNAPPs), a few examples: 1) https://doi.org/10.1038/nmicrobiol.2016.162, 2) https://doi.org/10.1002/adfm.202107341.
Response: We agree with the reviewer. In that view, we have included the suggested references in the manuscript, as follows on page 18, lines 636-655:
“A class of nanoparticles based on star-shaped peptide polymers consisting of lysine and valine residues is also used to treat bacterial infections. Termed as structurally nanoengineered antimicrobial peptide polymers (SNAPPs), they are highly stable compared to other polymers. SNAPPs in the form of 16- and 32-arm star peptide polymer nanoparticles showed a broad spectrum of activity against Gram-negative bacteria such as E. coli, P. aeruginosa, K. pneumoniae, A. baumannii with MBC values lower than 1.61 µM while demonstrating no significant cytotoxicity. The SNAPP 16 showed more than 99% of bacterial cell eradication in a mouse peritonitis model infected with A. baumannii, whereas only 20% of the control survived after 24 h. SNAPP 16 improves host cell innate immunity to A. baumannii in vivo by enhancing neutrophil infiltrate in the peritoneal cavity, while the control group shows no significant difference. According to the analyses performed, no microbial resistance was observed by colistin-resistant and MDR (CMDR) pathogens to the SNAPP since these nanoparticles presented different mechanisms of antibacterial action [70].
SNAPPs can also be applied for pulmonary delivery against respiratory bacterial infections such as pneumonia and tuberculosis. SNAPPs immobilized by different techniques in polyphenol-based capsules were internalized by alveolar macrophages in vitro. They were effective against E. coli with MIC values ​​of approximately 30 μg. mL−1 with sustained release and non-significant cytotoxicity. Furthermore, they remain stable in nebulized droplets. [71].”
- The article is also silent about antibody-bactericidal macrocyclic peptide conjugates (ABCs): for example, https://doi.org/10.1002/cbic.201800295. I would encourage the authors to include a discussion about such examples/possibilities.
Response: We agree with the reviewer. In that view, we have included the suggested reference in the appropriate section “Other types of nanomaterials” in the manuscript, as follows on page 24, lines 831-834:
“An antibody targeting E. coli was used to develop antibody–bactericidal macrocyclic peptide conjugate (ABCs), using the AMPs CAP-18, SMAP-29, and BMAP-27 from the cathelicidin family. The ABCs conjugates were effective against E. coli at nanomolar concentrations and had minimized hemolytic activity [72].”
A few minor typographical errors:
- Page 4, Line 130: “…monoclonal antibodies, were also conjugated with Au-NPs nanoparticles” – Unnecessary repetition can be removed.
Response: We agree with the reviewer. In that view, we have rewritten the sentence, as follows on page 6, lines 166-168:
“The peptide HuAL1, derived from complementarity-determining regions of monoclonal antibodies, was conjugated with Au-NPs and showed the …”
- Page 4, Line 139: “…the number of surviving HeLa cells was more than 3.6-fold treated with Lys AB2 P3-His-Au-NP-Apt after 24 h treatment.” – Please consider reframing/correcting the sentence.
Response: We agree with the reviewer. In that view, we have rewritten the sentence, as follows on page 6, lines 178-180:
“…LysAB2 P3-His-Au-NP-Apt increased the number of viable HeLa cells by more than 3.6-fold compared to the control. “
- Page 5, Line 192: “Nanoparticles (NPs) from derivatives…” – Unnecessary. Short form can be used as it has already been mentioned earlier.
Response: We agree with the reviewer. In that view, we have rewritten the sentence, as follows on page 8, line 271:
“NPs from derivatives…”
- Page 5, Line 192: “…derivatives of other metals like silicon…” – Silicon is neither metal nor non-metal; it is a metalloid.
Response: We agree with the reviewer. In that view, we have corrected the information on the sentence and included the respective reference, as follows on page 8, line 271:
“NPs from derivatives of metalloids like silicon [73] are...”
- Page 13, Line 583: “…standardized testing under in vitro and vivo conditions…” – Needs to be corrected; in vitro and in vivo.
Response: We agree with the reviewer. This sentence was removed.

Reviewer 4 Report
Comments and Suggestions for Authors
The review describes the therapeutic potential of antimicrobial peptide-based treatments for bacterial infections, with a focus on the nanomaterial delivery systems of antimicrobial peptides. It summarizes recent advances in antimicrobial peptide-nanoparticle conjugation methods based on various studies from the past five years. However, the manuscript requires major revisions before publication, and the following issues need to be addressed.
1. The review should include additional corresponding diagrams in the section summarizing the research progress on different antimicrobial peptide carriers to help readers better understand the content. Having only one illustration (Figure 1) in the text is far from sufficient for a review article.
2. The author should add tables comparing the advantages and disadvantages of various carriers and the strengths and weaknesses of different carriers in different biomedical applications.
3. The conclusion section lacks sufficient summary and discussion. The authors should provide their own insights on the challenges associated with antimicrobial peptides, along with potential solutions and unique perspectives on future development and advancements in the field.
4. In the section introducing the research progress of different antimicrobial peptide carriers, the paper merely lists a collection of related literature without offering the authors' own insights or critiques. Additionally, the reported advancements lack a strong logical connection, and there is no clear, cohesive review framework reflecting the authors' unique perspective.
5. Some reviews about classical nanosystems for drug delivery should be added in the introduction. (International Journal of Biological Macromolecules, 2023, 251: 126337; Extracellular Vesicles and Circulating Nucleic Acids, 2024, 5:344-57) Authors are suggested to refer to them.
Author Response
Dear Dr. Maresca and Dr. Simões,
Please find enclosed the response to the reviewer’s letter for the review entitled “Antimicrobial peptide delivery systems as promising tools against resistant bacterial infections” by Kamila B. S. Oliveira, Michel L. Leite, Nadielle T. M. Melo, Letícia F. Lima, Talita C. Q. Barbosa, Nathalia L. Carmo, Douglas A. B. Melo, Hugo C. Paes, and Octávio L. Franco that we are resubmitting to the Conference Special Issue "Rise of Antibiotic Resistance: Mechanisms Involved and Solutions to Tackle It” in the Antibiotics Journal. We appreciate and agree with the reviewers' suggestions. Original comments are in italics, and our answers are in regular typeface. All modifications are highlighted in the modified version of the manuscript. We believe the manuscript is improved after the changes suggested by the editorial board.
RESPONSES TO REVIEWERS’ COMMENTS
Reviewer 4. The review describes the therapeutic potential of antimicrobial peptide-based treatments for bacterial infections, with a focus on the nanomaterial delivery systems of antimicrobial peptides. It summarizes recent advances in antimicrobial peptide-nanoparticle conjugation methods based on various studies from the past five years. However, the manuscript requires major revisions before publication, and the following issues need to be addressed.
Major Comments:
1 The review should include additional corresponding diagrams in the section summarizing the research progress on different antimicrobial peptide carriers to help readers better understand the content. Having only one illustration (Figure 1) in the text is far from sufficient for a review
article.
Response: We agree with the reviewer. We agree that the review needs additional graphic elements. In that view, we added one more figure to the manuscript about the mechanism of action of AMPs conjugated with nanomaterials to help readers better understand the content of the review. Reviewer 1 also asked for this modification, so it is already described above in this letter, as follows on page 4. However, we disagreed that a diagram summarizing the research progress on different antimicrobial peptide carriers is necessary as we already have one figure summarizing different antimicrobial peptide carriers.
2 The author should add tables comparing the advantages and disadvantages of various carriers and the strengths and weaknesses of different carriers in different biomedical applications.
Response: We agree with the reviewer. In that view, since the focus of our review is the AMP delivery using nanomaterial carriers, we included a table comparing the strengths and weaknesses of different nanomaterial carriers for AMP delivery. Reviewer 1 had also requested the addition of new graphical elements, so the new table can be seen just above in this letter, as follows on pages 25-29.
3 The conclusion section lacks sufficient summary and discussion. The authors should provide their own insights on the challenges associated with antimicrobial peptides, along with potential solutions and unique perspectives on future development and advancements in the field.
Response: We agree with the reviewer's suggestion. In that view, we have rewritten the whole conclusion section describing our perspective associated with the challenges of AMP nanoformulation along with potential solutions and future perspectives. Reviewer 1 and 3 also asked for this modification, so the new conclusion section can be analyzed above in this letter, as follows on pages 33-34.
4 In the section introducing the research progress of different antimicrobial peptide carriers, the paper merely lists a collection of related literature without offering the authors' own insights or critiques. Additionally, the reported advancements lack a strong logical connection, and there is no clear, cohesive review framework reflecting the authors' unique perspective.
Response: We agree with the reviewer's suggestion. In that view, we have included our perspective throughout the manuscript, as follows on page 6, lines 169-172:
“The conjugation with Au-NPs showed antibacterial activity with lower concentrations than the peptide alone, which is very important since lower concentrations avoid side effects occurrences.”
as follows on page 6, lines 176-178:
“...achieving a survival rate of 70% of infected mice while in the control group treated with the peptide alone, the survival rate achieved was only 20%. [29].”
as follows on page 6, lines 192-194:
“These concentrations were lower compared to the free Ura 56, demonstrating an increased potency in the peptide efficacy. This may be possible because the NPs conjugation protects the peptide.”
as follows on page 6, lines 197-198:
“...probably due to the difficulty of protease to bind to the Ura-56 attached to the surface of AuNPs.”
as follows on page 7, lines 211-215:
“...while the MIC values for P-13 alone were much higher, implying a potent synergic effect on the bacterial activity. Moreover, Ag-NPs cytotoxicity was greatly reduced after being conjugated with the P-13 peptide, probably because the peptide covered the Ag-NP outer shell and reduced the metal surface contact with the cells. Besides, P-13-Ag-NPs were more selective toward bacterial cells than mammalian cells [74].”
as follows on page 7, lines 221-223:
“The conjugate allows better electrostatic interaction with the cell membrane of Gram-negative bacteria, generating better membrane permeability.”
as follows on page 7, lines 228-234:
“Tryasine-Ag-NPs were more effective than tryasine alone against S. aureus and E. coli at 30 and 28 µg. mL-1 MICs, respectively. These values are around 50% lower than peptide alone, probably due to the peptide interaction with the membrane of bacteria, which causes the increase of permeability, leading to the antibacterial effect of Ag-NP. Tryasine-AgNPs exhibited 1% hemolytic action on human erythrocytes. Therefore, the tryasine conjugation with Ag-NP is a promising candidate for bacterial infection with low toxicity [32].”
as follows on page, lines:
“...compared to free NZW peptide and maintained their effectiveness in an in vivo murine model infected in lungs with M.tuberculosis, achieving a reduction of 88% (Tenland et al., 2019).”
as follows on page 9, lines 315-321:
“Biofilms are very difficult to access with antimicrobial molecules because pathogenic cells are protected by thick and dense extracellular polymeric substances (EPS), which are composed of polysaccharides, proteins, glycoproteins, lipids, surfactants, and nucleic acids. This feature makes biofilms commonly resistant to multiple drugs [33]. Given this, these data demonstrate how the combination and synergy between AMP, antibiotics, and nanoparticles is relevant in the treatment of resistant infections.”
as follows on page 10, lines 334-335:
“The conjugation to MSNs provides better cellular uptake and antibacterial efficacy.”
as follows on page 10, lines 343-347:
“These studies demonstrate how the conjugation of AMPs with metal NPs has significant contributions to the treatment of infectious diseases. This formulation allows the distribution of AMP in vivo, reducing its cytotoxicity and protecting it against degradation by proteases AMPs conjugated with metal NPs provide reduced toxicity, greater antibacterial activity, superior targeting, and greater stability than free AMPs.”
as follows on pages 11-12, lines 379-383:
“...while MH5C-Cys-PEG (2kDa) did not show inhibition or eradication of the biofilms [76]. The exact reason why there is this weight dependency is unclear. This data reinforces the importance of a careful study of nanoparticles study and further design in combination with target AMP structure analysis.”
as follows on page 12, lines 394-398:
“Interestingly, the antibacterial activity of N6 PEGylated with more PEG lengths was reduced, possibly due PEGylation change in structural features of peptides.
Nevertheless, PEGylation with longer PEG spacer lengths enhanced the antibacterial properties of the peptide KR12.”
as follows on page 12, lines 405-406:
“These results point to the relevance of the individuality of each system, considering the peptide.”
as follows on page 13, lines 432-434:
“Here, PVA is adsorbed on the surface of PLGA and reduces the adhesion of hydrophobic PLGA to airway mucus.”
as follows on page 13, lines 439-441:
“In addition to extending and increasing the therapeutic effect against P. aeruginosa lung infections compared to AMPs in their free soluble form, the nanoformulation enabled airborne administration.”
as follows on page 13, lines 444-445:
“Here, the PLGA was used to enhance BMNPs internalization, since presents biocompatibility and biodegradability properties.”
as follows on page 14, lines 477-480:
“These studies demonstrate how the conjugation of AMPs with PLGA NPs has significant contributions to the treatment of infectious diseases, providing protection of AMP against degradation by proteases, long-lasting release of target AMP at the site of infection, and reduction of cytotoxicity.”
as follows on page 15, lines 516-519:
“.... more effective than a free peptide. It is important to highlight that the free capsule showed a mild cytotoxic effect on bacterial cells at the highest concentration, corroborating that chitosan has an antibacterial effect itself and can improve efficacy conjugated with an AMP.”
5 Some reviews about classical nanosystems for drug delivery should be added in the introduction. (International Journal of Biological Macromolecules, 2023, 251: 126337; Extracellular Vesicles and Circulating Nucleic Acids, 2024, 5:344-57) Authors are suggested to refer to them.
Response: We appreciate the comment reviewer. In that view, we have included the suggested review about extracellular vesicles at the end of the lipidic nanoparticles section as they are a potential solution for the endosome escape problem of the lipidic nanoparticles. Considering the review focus is AMPs delivered by nanomaterials, we have also included two other references about AMPs conjugated with the extracellular vesicles, as follows on page 23, lines 780-799:
“A problem that lipidic nanoparticles face in general, like other delivery vehicles, is endosome escape. Delivery vehicles can enter early endosomes and be either sent back to the plasma membrane or proceed to the lysosomal pathway, thus causing degradation of the delivery system. Extracellular vesicles (EVs) may be an alternative to this problem. EVs are membrane-bound particles naturally secreted by various cells and are subclassified as exosomes, microvesicles/ectosomes, and apoptotic bodies. They can package and transport various bioactive molecules, making them promising molecules as delivery systems. Among the molecules EVs transport are lipids, nucleic acids, and proteins [77]. In addition to AMPs being encapsulated by EVs, they can be delivered by surface modification coating them. Specific EVs were coated with a novel cationic AMP, AMP-A, with good antibacterial and biocompatibility properties. This change made the surface charge of the vesicles neutral. This physical change improved the antibacterial activity against E. coli, showing MBC values 2-fold lower than AMP-A alone. It also improved biocompatibility and reduced the peptide’s cytotoxic effect [78]. The data indicate that EVs are a potential alternative to improve the antibacterial activity and cytocompatibility of AMPs. Another EV type, a rose-derived exosome-like nanoparticle, encapsulated ELNs AMPs. The nanoconjugates promoted enhanced antibacterial activity against intracellular MRSA, 2.5 times greater than ELNs alone, in in vitro cell infection assays [79].”
We have included the other suggested review about poly(lactic acid) (PLA) at the end of the polymeric nanoparticles section as they are polymeric material. Considering the review focus is AMPs delivered by nanomaterials, we have also included one other reference about AMPs conjugated with PLA, as follows on page 18, lines 656-670:
“Another polymer used to deliver AMPs is poly(lactic acid) (PLA). This polymer has very favorable characteristics as a delivery system including biodegradability, compatibility with biomolecules and cells, and low production cost. Given this, PLA-based micro and nanofibers have been used for wound healing [80]. The AMP temporin L isolated from the skin of the frog Rana temporia has potent antibacterial activity with MIC values ranging from 0.3 to 3.6 µM for various bacterial strains [81]. Therefore, temporin L was conjugated to a cationic-based polymer for potential application in wound dressing. Initially, the temporin L peptide was functionalized with a polymer containing talcin and thiazolium groups, forming the peptide conjugate polymer (PTTIQ-AMP). This conjugate was subsequently incorporated into PLA electrospun fibers to analyze the synergic activity. The PTTIQ-AMP conjugated into PLA fibers has shown improved antibacterial performance been capable of reducing E. coli and E. faecalis cells to 99.999% compared to control groups. This result can be attributed to diffusion capacity and leaching proprieties contributing to the effective and sustainable release of temporin L [81].”

Reviewer 5 Report
Comments and Suggestions for Authors
In this review titled “Antimicrobial Peptide Delivery Systems as Promising Tools Against Resistant Bacterial Infections,” the authors have summarized recent advancements in the field of antimicrobial peptides utilizing various nanosystems. While the authors have done an excellent job summarizing different research articles, the review lacks sufficient discussion to infer how these nanosystems overcome the limitations of antimicrobial peptides. Please address the following major and minor comments to strengthen this review article.
Major Comments:
-
The authors have summarized the results from recent research articles well, but they have not thoroughly discussed how these nanosystems address the challenges faced by antimicrobial peptides. The results are presented without further inference or exploration of their impact.
-
Including a pictorial summary of the results from different research papers, either in the form of tables or copyright-permitted images, would significantly improve the manuscript. Visual representations can enhance the clarity and usefulness of the review for readers.
-
The manuscript frequently mentions bacterial infections without specifying the type (e.g., topical, local, systemic, or organ-specific) when discussing the results. Additionally, the review does not clearly distinguish between treatments for topical, oral, or systemic infections using nanoparticles.
-
Beyond the conclusion, the manuscript lacks a discussion on how each type of nanoparticle can guide future research in the field of antimicrobial peptides. Including this perspective throughout the review would greatly enhance its contribution to the field.
-
While the authors report that certain nanosystems demonstrate improved antibacterial activity, they do not explain the underlying reasons for this improvement. Specific areas have been pointed out in the minor comments, but this should be addressed consistently throughout the manuscript.
Minor Comments:
-
The manuscript should be carefully proofread to correct any spelling errors.
-
In line 158, the authors mention stability in rat serum. How long was the stability observed, and what impact might it have on the treatment?
-
In lines 113-115, the authors state that immobilizing antimicrobial peptides might increase the stability of nanoparticles. However, reducing regions of electrostatic interaction during antibacterial interaction could also reduce nanoparticle stability. Does the referenced data support this claim?
-
In lines 220-221, the authors mention improved results from the co-delivery platform compared to the control and free peptide but do not explain the reason for this improvement.
-
In lines 272-273, the authors suggest that longer spacer length and higher density lead to better antibacterial activity. Why do longer spacer lengths or higher densities contribute to enhanced antibacterial effects?
-
In line 287, please clarify why the nanoformulation was found to be less toxic.
Minor editing of English language required.
Author Response
Dear Dr. Maresca and Dr. Simões,
Please find enclosed the response to the reviewer’s letter for the review entitled “Antimicrobial peptide delivery systems as promising tools against resistant bacterial infections” by Kamila B. S. Oliveira, Michel L. Leite, Nadielle T. M. Melo, Letícia F. Lima, Talita C. Q. Barbosa, Nathalia L. Carmo, Douglas A. B. Melo, Hugo C. Paes, and Octávio L. Franco that we are resubmitting to the Conference Special Issue "Rise of Antibiotic Resistance: Mechanisms Involved and Solutions to Tackle It” in the Antibiotics Journal. We appreciate and agree with the reviewers' suggestions. Original comments are in italics, and our answers are in regular typeface. All modifications are highlighted in the modified version of the manuscript. We believe the manuscript is improved after the changes suggested by the editorial board.
RESPONSES TO REVIEWERS’ COMMENTS
Reviewer 5. In this review titled “Antimicrobial Peptide Delivery Systems as Promising Tools Against Resistant Bacterial Infections,” the authors have summarized recent advancements in the field of antimicrobial peptides utilizing various nanosystems. While the authors have done an excellent job summarizing different research articles, the review lacks sufficient discussion to infer how these nanosystems overcome the limitations of antimicrobial peptides. Please address the following major and minor comments to strengthen this review article.
Major Comments:
1 The authors have summarized the results from recent research articles well, but they have not thoroughly discussed how these nanosystems address the challenges faced by antimicrobial peptides. The results are presented without further inference or exploration of their impact.
Response: We agree with the reviewer's suggestion. The other reviewers asked for the same modifications. Therefore, the discussion about how these nanosystems address the challenges faced by antimicrobial peptides is already described above.
2 Including a pictorial summary of the results from different research papers, either in the form of tables or copyright-permitted images, would significantly improve the manuscript. Visual representations can enhance the clarity and usefulness of the review for readers.
Response: We agree with the reviewer's suggestion. Reviewer 1 asked for the same modifications and the new tables regarding the results summary from the research papers described in the manuscript are already described above, as follows on pages 4, 10-11, 19-20 and 25-29.
3 The manuscript frequently mentions bacterial infections without specifying the type (e.g., topical, local, systemic, or organ-specific) when discussing the results. Additionally, the review does not clearly distinguish between treatments for topical, oral, or systemic infections using nanoparticles.
Response: We agree with the reviewer's suggestion. Therefore, we have included a table describing the common administration type for each AMP delivery system. Reviewer 1 asked for tables, so it is already described above, as follows on pages 25-29.
4 Beyond the conclusion, the manuscript lacks a discussion on how each type of nanoparticle can guide future research in the field of antimicrobial peptides. Including this perspective throughout the review would greatly enhance its contribution to the field.
Response: We appreciate the reviewer's comment. In that view, we have rewritten the whole conclusion section describing our perspective associated with the challenges of AMP nanoformulation along with potential solutions and future perspectives. Reviewer 1 and 3 also asked for this modification, so the new conclusion section can be analyzed above in this letter, as follows on pages 33-34.
5 While the authors report that certain nanosystems demonstrate improved antibacterial activity, they do not explain the underlying reasons for this improvement. Specific areas have been pointed out in the minor comments, but this should be addressed consistently throughout the manuscript.
Response: We appreciate the reviewer's comment. We would like to point out that these modifications regarding the reasons for improvement in activity performance, and reduction in cytotoxicity, among others, were made throughout the manuscript to improve the quality of the manuscript. The reviewers 3 and 4 asked for the same modification. Therefore, the reasons for the improvement are addressed consistently throughout the manuscript and is already described above on this letter.
Minor Comments:
1 The manuscript should be carefully proofread to correct any spelling errors.
Response: We appreciate the reviewer’s comment. In that view, the manuscript has been proofread to correct the spelling errors.
2 In line 158, the authors mention stability in rat serum. How long was the stability observed, and what impact might it have on the treatment?
Response: We appreciate the reviewer's comment. In that view, we have rewritten the sentence adding the requested information, as follows on page 6, lines 196-197:
“Furthermore, Ura56-PEG-Au-NPs demonstrated stability in rat serum while 45% of the free Ura 56 was degraded just 6h after incubation.”
3 In lines 113-115, the authors state that immobilizing antimicrobial peptides might increase the stability of nanoparticles. However, reducing regions of electrostatic interaction during antibacterial interaction could also reduce nanoparticle stability. Does the referenced data support this claim?
Response: We appreciate the reviewer's comment. In that view, we have rewritten the sentence correctly as follows on page 5, lines 142-145:
“AMPs have functional groups (amino, carboxyl, and thiol) with high affinity for gold or silver atoms, which can immobilize peptides by electrostatic and hydrophobic interaction [55] and reduce the toxicity of the metal nanoparticle and at the same time increase the AMP activity [54].”
4 In lines 220-221, the authors mention improved results from the co-delivery platform compared to the control and free peptide but do not explain the reason for this improvement.
Response: We appreciate the reviewer's comment. In that view, we have rewritten the sentence correctly as follows on pages 7-8, lines 242-244:
“Probably the AMP was responsible for the cytotoxicity reduction of AgNPs, covering the AgNPs shell and reducing their contact with the cells [13].”
5 In lines 272-273, the authors suggest that longer spacer length and higher density lead to better antibacterial activity. Why do longer spacer lengths or higher densities contribute to enhanced antibacterial effects?
Response: We appreciate the reviewer's comment. In that view, we have rewritten the sentence correctly as follows on page 12, lines 404-405:
“....since the long PEG spacer facilitates theKR12 access to the bacterial cell membrane [36].
6 In line 287, please clarify why the nanoformulation was found to be less toxic.
Response: We appreciate the reviewer's comment. In that view, we have rewritten the sentence correctly as follows on page 12, lines 427-429:
“These data indicate that PLGA particles can indeed reduce the toxicity of AMPs, probably because PLGA precisely delivers AMP to its site of action, preventing its interaction with macrophage cells.”
We would like to point out that these modifications regarding the reason why there was an improvement in activity performance, and reduction in cytotoxicity, among others, were made throughout the manuscript to improve the quality of the manuscript.

Round 2
Reviewer 1 Report
Comments and Suggestions for Authors
Authors should reduce the overall plagiarism of the manuscript before making final submission for publication.
Reviewer 3 Report
Comments and Suggestions for Authors
The authors have answered most of my queries. They have done quite a bit of additional writing with appropriate literature support in response to the raised concerns. As a result, the quality of the article after the revision, has improved significantly. I, therefore, strongly recommend the acceptance of the article for publication in the Antibiotics journal.
Reviewer 4 Report
Comments and Suggestions for Authors
The author figured out all the issues. I'd like to recommend acceptance for publication.
Reviewer 5 Report
Comments and Suggestions for Authors
The authors' efforts in revising the manuscript are greatly appreciated, and I agree with all the changes they have made.